



# Seismicity related to the eastern sector of Anatolian escape tectonic: the example of the 24 January 2020 Mw 6.77 Elazığ-Sivrice earthquake

**Mohammadreza Jamalreyhani[1,2], Pınar Büyükakpınar[1,3], Simone Cesca[1], Torsten Dahm[1], Henriette Sudhaus[1,4], Mehdi Rezapour[2], Marius Paul Isken[1,4], Behnam Maleki Asayesh[5], and Sebastian Heimann[1]**

[1]GFZ German research centre for geosciences, Potsdam, 14473, Germany.
[2]Institute of geophysics, University of Tehran, 14155-6466, Iran.
[3]Regional Earthquake Tsunami Monitoring Center, Kandilli Observatory and Earthquake Research Institute, Boğaziçi University, Istanbul, 34684, Turkey.
[4]Department of Geosciences, Christian-Albrechts-University Kiel, Kiel, 24118, Germany.
[5]International Institute of Earthquake Engineering and Seismology, Tehran, 19537-14453, Iran.

*Correspondence to*: Mohammadreza Jamalreyhani (m.jamalreyhani@gmail.com)

**Abstract.** The 24 January 2020 Mw 6.77 Elazığ-Sivrice earthquake (Turkey), responsible for 42 casualties and ~1600 injured people, is the largest earthquake affecting the East Anatolian Fault (EAF) since 1971. The earthquake partially ruptured a seismic gap. The mainshock was preceded by two foreshocks with Mw ≥ 4.9 and small seismicity clusters

occurring in the previous months close to the nucleation point of the main rupture. The significant aftershock sequence comprises twelve earthquakes with Mw≥4.5 within 60 days. We jointly model quasi co-seismic static surface displacements from Interferometric Synthetic Aperture Radar (InSAR) and high-frequency co-seismic data from seismological networks at local, regional and teleseismic distances to retrieve source parameters of the mainshock. We reconstruct the rupture process using a Bayesian bootstrap based probabilistic joint inversion scheme to obtain source parameters and their uncertainties.

Full moment tensor for 18 fore-/after-shocks with Mw≥4.3 are obtained based on the modeling of regional broadband data. The posterior mean model for the 2020 Elazığ-Sivrice mainshock shows that the earthquake, with a magnitude Mw 6.77, ruptured at shallow depth (5±2 km) with a left-lateral strike-slip focal mechanism, with a dip angle of 74°±2° and a causative fault plane strike of 242°±1°, which is compatible with the orientation of the EAF at the centroid location. The rupture nucleated in the vicinity of small foreshock clusters and slowly propagated towards WSW, with a rupture velocity of

~2100±130 m s$^{-1}$ and ~27 s rupture duration. The main rupture area, with a length of ~26±5 km, only covered 70% of the former seismic gap, leaving a smaller, unbroken segment of ~30 km length to the SE with positive stress change. The subsequent aftershock sequence extended over a broader region of ~70 km in length, spreading to both sides of the mainshock rupture patch into the regions experiencing a stress increase according to our Coulomb stress modeling. Our results support the hypothesis of a shallow locking depth of the Anatolian micro-plate, which has a possible implication to

the seismic bursts along the EAF and alternating seismic activity on the North Anatolian and the East Anatolian faults.



## 1 Introduction

The East Anatolian Fault (EAF), striking SW-NE over a length of ~550 km, is one of the most seismically active tectonic structures in eastern Turkey (McKenzie, 1976). The regional seismotectonic setting is controlled by the collision of the

Arabian and Eurasian plates at a rate of ~15 mm yr$^{-1}$ (Reilinger et al., 2006). Since the middle Miocene (~15 Ma years ago), the Anatolian continental block in between the two plates escapes westwards by ~20 mm yr$^{-1}$ (Reilinger et al., 2006; Philippon et al., 2014), where both the East Anatolian Fault (EAF) and the North Anatolian Fault (NAF) accommodate most of the tectonic displacement. Recent GNSS studies showed that the slab pull forces acting beneath western Turkey along the Hellenic and Cyprus arcs play a significant role in the SW motion of the Anatolian block escape tectonics (McClusky et al.,

2000; Reilinger et al., 2006). The mantle flow beneath western Arabia supports the westward motion of the Anatolian lithosphere (Wei et al., 2019). The EAF experienced many devastating earthquakes (Mw>7) over historical times (Fig. 1), causing severe fatalities and extensive damage (Ambraseys, 1989; Ambraseys and Jackson, 1998; Nalbant et al., 2002; Duman and Emre, 2013). However, differently from the NAF, it did not host major earthquakes during the last hundred years (Fig. 1): the most recent, large earthquake on the EAF dates to 1971 (Mw 6.7 Bingöl earthquake, Duman and Emre,

2013). A more recent strong earthquake along the EAF was the 2010 Mw 6.1 Kovancilar earthquake (Tan et al., 2011). Recurrence time of large earthquakes on the EAF is estimated at ~190 years in an analysis of a 3800-year record of lacustrine sediments in the Lake Hazar (Hubert-Ferrari et al., 2020), adjacent to the 2020 Elazığ-Sivrice earthquake. Paleo-seismic studies indicate that the NAF and EAF had highly variable recurrence times from decades to centuries during the last millennium, and that seismic activity is alternating between the two faults (e.g. Agnon et al., 2006). A shallow locking depth

(Cavalié and Jónsson, 2014) may facilitate an alternating rotational motion of the Anatolian block and be a reason for transient hazard at the EAF and NAF. The EAF has been the target of many seismological studies. Taymaz et al. (1991) first estimated an average slip rate of ~29 mm yr$^{-1}$ along the EAF during the last century. The GNSS measurements show that the present-day slip rate across the EAF is only 8-11 mm yr$^{-1}$ (McClusky et al., 2000; Reilinger et al., 2006). Comparing present day seismicity and historical seismic activity, inferred from paleoseismological and historical studies, Çetin et al. (2003)

suggest fault locking and accumulation of elastic strain along the entire EAF. Cavalié and Jónsson (2014) and Bletery et al. (2020) proposed heterogeneous and shallow (~5 km) locking depth for the EAF, based on InSAR timeseries and GNSS data. The EAF is nowadays considered as a segmented and structurally complex fault system, with a dominant left lateral strike-slip mechanism (Duman and Emre, 2013). However, while tectonic stress is mostly accommodated by strike-slip faulting (Örgülü et al., 2003), some variability in the earthquake focal mechanisms (Taymaz et al., 1991) is observed (e.g. SW of

Karliova junction, eastern Anatolian Plateau). Bulut et al. (2012) characterize the EAF as a left-lateral strike-slip system, involving NE-SW and EW oriented segments which run parallel to the segmented trend of the main fault. Besides the dominant strike-slip mechanisms, Bulut et al. (2012) found evidence for additional thrust faulting on EW trending structures and normal faulting on NS trending secondary faults. Duman and Emre (2013) proposed seven main segments along the southern strand of the EAF and suggested different seismic gaps. The largest one, with a length of almost 100 km, is the



Pütürge segment (Fig. 1), which strikes N60°E and extends between the Hazar Lake and the Yarpuzlu double bend (Duman and Emre, 2013). An 11 km left-lateral fault offset since the Pliocene was estimated by Duman and Emre (2013), based on the offset of the Firat River, which crosses the EAF within the Pütürge segment. Duman and Emre (2013) identified this segment as a seismic gap and as a candidate to host large earthquakes. This gap was partially filled by the 2020 Mw 6.77 Elazığ-Sivrice earthquake, which struck in the NE part of the Pütürge segment. Bletery et al. (2020) by assuming the slip rate

of ~12 mm yr$^{-1}$, showed that this earthquake released the moment accumulated over 221±26 years which suggests a recurrence time between 200 and 259 years in the Pütürge segment. This inference is supported by paleoseismic studies, finding a mean recurrence time of ~190 years (Hubert-Ferrari et al., 2020). Historical and paleoseismic records further suggest that faulting along the EAF is occurring in seismic bursts, sequences similar to the 19th century earthquakes, characterized by five events with Mw 6.7 to 7 in 1866, 1874-1875, 1893 and 1905 (Ambraseys, 1989; Hubert-Ferrari et al.,

75 2020).

The 2020 Mw 6.8 Elazığ-Sivrice earthquake is the largest earthquake on the EAF for more than a century, and one of the best monitored to date. According to the report of Turkey's Disaster and Emergency Management Authority (AFAD), 42 people were killed and more than 1600 were injured in this earthquake (AFAD report, 2020). Proposed source mechanisms for the Elazığ-Sivrice earthquake (e.g. U.S. Geological Survey-National Earthquake Information Centre (USGS-NEIC),

Global Centroid Moment Tensor (GCMT) and GFZ German Research Center for Geosciences (GEOFON) indicate a left-lateral slip on a near-vertical WSW striking plane, with a shallow depth of 8-20 km (Table 1). Besides the mainshock, two M 4.0+ foreshocks and 31 M 4.0+ aftershocks occurred at a shallow depth of ~3-18 km (DDA Catalog, last accessed March 31, 2020) along the Pütürge segment. Most of the foreshocks took place in the vicinity of the Elazığ-Sivrice earthquake hypocenter. Aftershocks aligned NW of the EAF surface trace and spread over a larger segment of ~70 km in length. In this

work, we model the rupture process of the Elazığ-Sivrice earthquake and its largest fore- and aftershocks, by joint inversions of seismological and deformation data. We calculate the imparted stress change caused by this earthquake on the surrounding area and the relation between triggered aftershocks and stress changes. We discuss the rupture process with respect to fault segmentation, locking depth and hazard implications of a destructive earthquake in the Pütürge seismic gap along the EAF.

**2 Method**

In this study, we jointly use seismological and static surface displacement data to model the Elazığ-Sivrice mainshock. Furthermore, we use seismological data for the largest earthquakes in its sequence. We use a combination of regional and teleseismic data to retrieve earthquake point source parameters for the mainshock and, later, a combination of near-field strong motions along with InSAR surface deformation, to retrieve a kinematic finite source model. Moment tensor inversion

for weaker events is based on broadband data at regional distances. For all events we apply the probabilistic earthquake source inversion framework Grond (Heimann et al., 2018). By implementing a Bayesian bootstrap approach, the Grond optimization explores the full model space and maps model parameter uncertainties and trade-offs. Synthetic seismograms



are computed using pre-calculated Green's functions (Heimann et al., 2019) for seismic waveform and static displacement at local, regional and teleseismic distances based on a local crustal model (Acarel et al., 2019, Fig. S1), the CRUST2.0 model

for the target region (Bassin et al., 2000) and a global AK135 model (Kennett et al., 1995), respectively. Additional details for each used methodology are provided in the Pyrocko webpage (pyrocko.org — pyrocko.org).

**3 Mainshock**

Seismic records of the mainshock were recorded with a good signal-to-noise ratio (SNR) by broadband sensors at teleseismic and regional distances, except below ~300 km, where data are often saturated. Thus, we estimate the full moment tensor,

based on the inversion of regional and teleseismic data. We model long period (20-100 s) teleseismic P and SH body waves at 76 stations, homogeneously distributed at a distance of 30-80 degrees (Fig. S2); simultaneously, we model regional full waveforms for broadband seismic stations (Fig. S4) of the Kandilli Observatory and Earthquake Research Institute (KOERI) and GEOFON seismic network in a distance between 350 and 950 km. Regional full waveform observations have been revised manually to exclude noisy and saturated traces and have been restituted to displacement. Modeling has been

performed in the frequency band 0.008-0.03 Hz, representing dominant periods of the Rayleigh and Love waves. The results of moment tensor inversions, both, based on either the joint dataset or separately on the teleseismic datasets, are available in Table 1. Our results for the focal mechanism based on joint inversion of teleseismic and regional datasets confirm that the causative fault plane for the Elazığ-Sivrice mainshock has a strike of 246°±3° (assuming ENE-WSW fault orientation) and a dip of 70°±9°, with a left-lateral strike-slip mechanism (rake -16°±9°), within 68% of confidence. All fault plane angles are

very well resolved with uncertainties not exceeding 9°. We also resolved a shallow centroid depth of 5±2 km and a moment magnitude Mw 6.77±0.1. The centroid is located ~8 km WSW of the hypocenter, which suggests a rupture propagation towards WSW.

To perform the finite source optimization/inversion, we rely on near-field data. We used 6 local strong motion stations (Fig. 2, epicentral distances < 95 km) of the AFAD network, fitting velocity records in the frequency band 0.08-0.20 Hz. The

largest peak ground acceleration (PGA) and peak ground displacement (PGD), recorded at the station 2308, in the city of Sivrice, are 2.93 m s$^{-2}$ and 0.11 m, respectively (AFAD, 2020). The surface displacement caused by Elazığ-Sivrice earthquake was measured using Sentinel-1 A/B satellites data, from the European Space Agency (ESA), both for ascending and descending orbits and a temporal baseline of 6 days. For the interferometry we used the SAR images acquired from ascending orbits on January 21 and January 27, and the acquisitions from descending orbits on January 22 and January 28.

For the SAR image processing and the differential SAR interferometry we employed the SNAP software from ESA, with the implemented interferometric phase unwrapping through SNAPHU (Chen and Zebker, 2002). The postprocessing of the displacement maps was done using the software toolbox Kite (Isken et al., 2017). Here, we applied irregular quadtree data subsampling (Jónsson et al., 2002) and empirical variance-covariance estimation of the data error, which is used for data weighting in the optimization (Sudhaus and Jónsson, 2009). The wrapped interferograms show a maximal line-of-sight

displacement on the northern side of the fault (Fig. S3). No significant surface faulting has been observed in the coseismic





interferograms. Some surface cracks, rockfalls, landslides, and liquefaction were reported (Lekkas et al., 2020). We estimated a rupture length of 26±5 km and a rupture width of 9±1 km, with a uniform slip of 1.6±0.2 m, within 68% of confidence. The causative fault plane has a strike of 242°±1°, dip of 74°±2° and rake of 0°±2° (Table 1). The finite inversion provides evidence for a slow, shallow, unilateral rupture. The estimated rupture velocity (~2100±130 m s$^{-1}$) leads to an

overall rupture duration of ~27 s. This rupture velocity is almost at the lower range of velocities in the global catalog (Chounet et al., 2018). An independent analysis of apparent rupture duration (Cesca et al., 2011) for the mainshock, confirmed the azimuthal pattern of apparent durations (Fig. S5), which denotes unilateral ruptures. The rupture directivity, modulates the point source radiation pattern, producing higher amplitude and higher frequencies in front of the rupture and the opposite patterns in the opposite direction, explains the observed spatial distribution of saturated seismic stations (Fig.

S4): SW of the ruptured area, broadband data are saturated even at more than 500 km epicentral distance. Moreover, the azimuthal pattern for the PGA of strong motion stations supports the WSW directivity (Fig. 2). Similarly, it partially explains why most damages have been reported at the Malatya and Adiyaman cities (AFAD, 2020), which are located W-SW of the epicenter.

**4 Seismic sequence**

Locations and focal mechanisms of the foreshock-aftershock sequence are useful to understand the process of mainshock nucleation and the triggering of aftershock activity upon the mainshock. In addition, this information reconstructs the geometry of the EAF along the activated segment and the distribution of moment release over time. We rely on the AFAD catalog (DDA Catalog) to evaluate the spatial and temporal distribution of seismicity since January 2019. The catalog lists 80 foreshocks and 1200 aftershocks above magnitude Ml 2.0 and ~3700 events with Ml larger than 1 (~2000 aftershocks

with Ml 1+ and azimuthal gap < 120°, last accessed March 31, 2020). This dense seismic network has good azimuthal coverage (azimuthal gap <60°), and epicentral location uncertainties are estimated below 2 km (DDA Catalog). The aftershocks relocations by AFAD confirm the spatial distribution of the original absolute locations. It is noteworthy that the seismicity rate along the Pütürge segment has been relatively high in recent years. This segment hosts moderate magnitude earthquakes more than the neighboring segments (Fig. 1). Two small clusters have been located in the vicinity of the

mainshock hypocenter in the months preceding the earthquake (Last activity before 2019 along the Pütürge segment with Mw ~5 was in 2012, DDA Catalog). These clustered activities accompanied the 4 April 2019 Mw 5.2 and 27 December 2019 Mw 4.9 earthquakes, located ~5 km NE and N of the Elazığ-Sivrice mainshock hypocentre, respectively (Fig. 3). Both earthquakes were followed by small aftershocks. Most of the foreshocks including two Mw ~5 are located very close to the mainshock nucleation point, suggesting that they could have played a role in the mainshock preparation (Abercrombie and

Mori, 1996; Lippiello, 2012). The aftershocks are mostly clustered at three main areas (Fig. 3). One cluster is located in the east of the fault plane (vicinity of the Hazar Lake), another small cluster is located also east, but close to the epicenter of the mainshock, and a third cluster is located at the western end of the ruptured plane. All aftershocks are located north of the surface fault trace, confirming the NNW dip of the fault plane. In addition, the different spatial extents of aftershocks in two





parts of the ruptured segment may suggest a fault segmentation along the Pütürge segment, with sub-segments with different dip angles. Only a few aftershocks occurred in the main asperity area; this is in agreement with Das and Henry (2003), who suggested that generally few aftershocks occur in the high slip region of the fault. The spatial distribution of aftershocks shows a clear gap, which matches spatially the main asperity region, with an area of ~25×10 km, corresponding to the geometry of the main ruptured area, as resolved by our finite fault modeling of the mainshock.

We performed a moment tensor inversion for 18 earthquakes (2 foreshocks and 16 aftershocks) with Ml ≥ 4.3. For this purpose, we proceed as for the point source inversion. However, due to the weaker magnitude, we rely on only regional broadband data of the KOERI network (Fig. S3). We model full waveforms and amplitude spectra in the frequency band 0.02-0.05 Hz. Focal mechanisms solution of the foreshocks and aftershocks show mostly strike-slip mechanisms with a depth range of ~3-18 km (Fig. 3). The foreshocks and aftershocks dips indicate NNW dipping fault planes (with a broad distribution of dip angles in the range of ~23°-85°). The details of focal mechanisms solutions, compared to those reported by other agencies, are available in Table S1. The eastern part of the seismic activity (Fig. 3, c3) mostly displays strike-slip mechanisms, similar to the mainshock, and a depth range of 3-18 km. One shallow normal mechanism may have occurred due to local stress change induced by the mainshock. The central part of the activity (Fig. 3, c2), including two large foreshocks that are located close to the mainshock hypocenter, also shows strike-slip mechanisms; centroid depths are here shallower, and similar to the depth of the mainshock. The spread distribution of seismicity in the western part (Fig. 3, c1), which is located at the end of the rupture, may indicate fault complexity and/or activation of other local faults. Low dip angles of the deepest events suggest that the fault has here a more gentle dip angle in its deeper part.

**5 Coulomb failure stress change analysis**

In the last decades many studies confirmed the role of earthquake interactions, where the occurrence of an earthquake may influence the time and locations of future events by modifying the state of stress in the nearby faults (Stein et al., 1992; Hainzl et al., 2010; Asayesh et al., 2018). In this study, we used the software Coulomb 3.3 (Lin and Stein, 2004; Toda et al., 2005) to calculate the coseismic static stress changes for the Elazığ-Sivrice earthquake. We assume Young modulus, shear modulus, and a Poisson ratio equal to $8×10^4$ MPa, $3.2×10^4$ MPa, and 0.25, respectively. Moreover, we utilize the middle value of the apparent coefficient of friction equal to 0.4 for ordinary faults (King et al., 1994; Parsons et al., 1999). We assumed the Earth as a homogeneous elastic half-space and the causative fault of the Elazığ-Sivrice earthquake is considered as a rectangular dislocation (26 km long 9 km wide with mean slip equal to 1.8 m). By considering the EAF as a receiver fault and a depth of 5.5 km, our results show lobes of positive shear stress change in the prolongations of the rupture area (Figs. 4a, S6). We also observe the off-fault lobes of positive stress change on both sides of the fault (Fig. 4a; high-stress region perpendicular to the fault). These lobes are due to normal stress changes which may increase off-fault failures. In order to investigate the correlation between Coulomb failure stress changes and aftershock distribution, we calculate stress due to the Elazığ-Sivrice mainshock on optimally oriented faults (planes on which aftershocks might be expected to occur).





We consider the angle of greatest compression of the regional tectonic stress field equal to the N21°E (P axis of the mainshock mechanism). Our calculation demonstrates that the majority of about 1900 of aftershocks (Ml 1+ and azimuthal gap < 120°) are located in areas that received positive stress changes due to the slip on the mainshock rupture plane (Fig. 4b). Thus, the geometry of positive stress change lobes can also partially explain why aftershocks are more spread off the western part of the activated zone of the fault, compared to the eastern part.

## 6 Discussion

The Elazığ-Sivrice earthquake ruptured with strike-slip mechanism along an ENE-WSW fault, steeply dipping (~ 74°) to the NNW, which is compatible with the orientation of the EAF at the Pütürge segment (Fig. 3). The main ruptured area, with a length of ~26 km and a rupture area of ~250 km$^2$, propagated unilaterally toward the WSW with a duration of 27 s, at a speed of ~2100 m s$^{-1}$. This low rupture velocity can be explained by the shallow centroid depth (~5 km) and low shear-wave velocity in the upper crust. The mainshock started to nucleate from the topper part of the fault plane (Fig. 3b). This observation conflicts with a tendency reported by Mai et al. (2005) for the most strike-slip earthquakes (nucleation near the bottom). The uppermost edge of the rupture is modeled at ~2.5±0.3 km depth, which explains the lack of surface rupture. Our uniform-slip finite source model is in general agreement with the preliminary model published by the USGS based on teleseismic body and surface wave data, except for the dip angle and centroid depth (Table 1). The USGS source time function (STF) shows two-moment release pulses at ~10 and ~15 s after the origin time. Our simplified source model (uniform slip) can not resolve multiple STF pulses; however, these are not easily discernible in the local acceleration data. Our finite source model for the Elazığ-Sivrice earthquake suggests that this earthquake broke a shallow asperity, compatible with the shallow locking depth (~5 km) suggested for the EAF by Cavalié and Jónsson (2014). The Elazığ-Sivrice earthquake caused positive stress changes at both segments of the EAF at the NE and SW ends of the ruptured plane (Figs. 4a, S5). Both segments loaded by positive Coulomb stress change experienced strong earthquakes in historical times (1874 and 1875 in the NE neighboring segment, 1905 at the SW segment, Duman and Emre, 2013). These loaded stresses can expedite future large earthquakes on either one of these segments.

The spatial distribution of aftershocks is in good agreement with the mainshock finite fault model. Aftershocks epicenters align along a narrow ENE-WSW band, bounded to the South by the EAF surface trace. However, the aftershock density is much larger at the eastern and western edges of the affected segment, with lower aftershock activity in a central region of ~25 km length, which matches the location and extent of the mainshock rupture area (Fig. 3). Such a spatial pattern of aftershocks has been observed for several earthquakes, in different tectonic settings (Dreger, 1997; Das and Henry, 2003). However, the clear rupture directivity has not played a role in producing a larger number of aftershocks ahead of the main rupture direction, as observed for other unilateral rupture earthquakes (Gomberg et al., 2003; Nissen et al., 2019). The location of almost all aftershocks north of the fault trace confirms the fault plane dip towards NNW.

The spatial extent of the aftershock cloud north of the EAF surface trace is peculiar: the seismicity is more laterally spread in the western segment compared to the eastern one. This observation can be explained by different scenarios: (1) the branching





of the EAF or the presence of multiple sub-parallel faults along the fault, (2) a change in the fault dip angles of the EAF,

steeper on the eastern side and mainshock rupture area and with smaller dip angle on the western side, (3) a deeper extent of the seismogenic volumes in the West or (4), a combination of these scenarios.

The first hypothesis, segmentation of the fault in sub-parallel branches with right-lateral step-overs is shown in the mapped fault trace by Basili et al. (2013). These step-overs set off the segments by only about a kilometer, however. Such segmentation can not be excluded from the aftershock distributions, which may not reach the required resolution, also

considering the known presence of secondary fault next to the EAF (Bulut et al., 2012). In the InSAR data the highest surface displacement gradients, close to the mapped fault trace and between the displacements towards and away from the radar, show a right-lateral offset between the south-west part and the north-east part of the fault of 1 km to 2 km (Fig. S3). The highest displacement gradients align very well with the mapped fault trace by Basili et al. (2013) and suggest a good agreement with fault activation if the fault dip is steep. However, scenario (2), a change in fault dip along-strike, without

change of depth of the upper boundary of fault slip, would also influence the location of the highest surface displacement gradients and the gradient value. A shallower fault dip in the southwestern part would reduce the highest displacement gradient and shift the gradient maximum towards east. In oblique line-of-sight data like InSAR displacement maps this is hard to verify, however. Scenario (3) is unlikely, because the depths estimated by earthquake relocation and aftershock centroid MT inversion do not change significantly along the fault. Both scenarios, 1 and 2 are plausible and not in

contradiction with the data. Some systematic residuals in the near-fault InSAR results of the finite slip modelling (Fig. 2) may point to a slight segmentation, but the overall good data fit in the single-segment finite fault modelling suggests that segmentation is not a first-order feature.

Scenario (2) could imply change of the dip angle at the western edge of the mainshock rupture area, separating the rupture area to the western aftershock domain. Such geometrical features could explain the termination of the mainshock rupture and

the preservation of a smaller seismic gap west of the Elazığ-Sivrice earthquake. The induced stress change due to the slip of Elazığ-Sivrice earthquake on optimally oriented faults shows an agreement of the positive lobes of Coulomb stress changes and the aftershocks distribution (Fig. 4b) and suggests that the majority of aftershocks are triggered by the mainshock. A few aftershocks are located in areas of negative stress change. This issue can be explained by the uncertainties in the applied fault slip model (Woessner et al., 2012) and parameters of the Coulomb equation. Moreover, considering secondary stress changes

due to the aftershocks themselves would render the pattern of stress changes more complex potentially reducing the number of aftershocks in the negative stress change areas (Asayesh et al., 2020).

Upon the occurrence of the Elazığ-Sivrice earthquake, we can divide the former 96 km Pütürge segment seismic gap into 3 domains (Fig. 3): (1) from Hazar Lake to the Elazığ-Sivrice earthquake hypocenter (~35 km), mostly affected by the aftershock sequence, (2) the mainshock rupture area and the region west of it that is affected by a spatial cluster of

aftershocks at the end of the rupture (~30 km), and (3) an unbroken ~30 km seismic gap, left between the region affected by the Elazığ-Sivrice earthquake sequence and the restraining Yarpuzlu double bend (Duman and Emre, 2013). This part of the Pütürge segment shows very little activity (Mw 3+) compared to the other part of the segment during 2007-2011 (Bulut et





al., 2012). This smaller seismic gap remains as a candidate to host large future earthquakes: considering its length, it could host an earthquake with a comparable magnitude as the Elazığ-Sivrice earthquake. The increased stress in this part of the
fault can expedite large earthquake activity in this region (Fig. 4a).

Thanks to the extended seismic monitoring in the last decades, we can observe and analyze seismicity patterns during the preparation phase of the Elazığ-Sivrice earthquake. The seismic activity with moderate magnitude in the last 120 years was higher along the Pütürge segment (Fig. 1, Histogram), compared to the neighboring EAF segments, which was affected by large earthquakes in the last centuries. However, the background seismicity at the Pütürge segment was characterized by
weak to moderate magnitude earthquakes only. This observation, alone, confirmed the seismic activity along this segment of the EAF; combined with the knowledge of a long-lasting seismic gap, indicating the potential for large earthquakes in this region. In the short time scale (months before the mainshock), we detected two small-scale spatial clusters of seismicity with moderate magnitude (4 April 2019 Mw 5.2 and 27 December 2019 Mw 4.9), which can be considered as foreshock of Elazığ-Sivrice earthquake, due to their vicinity to the mainshock hypocenters, their occurrence time, and the similarity of
their focal mechanisms with the mainshock (Fig. 3). The former moderate magnitude earthquake along the Pütürge segment occurred in 2012.

**7 Conclusion**

The Elazığ-Sivrice earthquake as an example of recent activity in the eastern sector of the Anatolian escape tectonic has ruptured the EAF along the Pütürge segment, filling a large part of the former Pütürge seismic gap. Thanks to the dense local
and regional seismic network, we were able for the first time, to accurately observe the foreshocks and aftershocks sequence of a large earthquake along the EAF. By joint inversion of local, regional and teleseismic data with InSAR data we could reconstruct the finite rupture process of the mainshock and retrieve MTs for the largest foreshocks and aftershocks.

The seismicity in the Pütürge segment had in the last years a higher rate than neighboring segments. Local seismicity clusters, appearing months prior to the Elazığ-Sivrice earthquake occurrence, probably track the slip instability onset. The
Elazığ-Sivrice earthquake nucleated in the vicinity of these clusters, but the rupture slowly migrated westward at shallow depth, breaking a locked asperity of ~26 km in length. The earthquake was followed by a large aftershock sequence, which affected a longer segment of the EAF. Only a few aftershocks occurred in the main asperity area and most of the aftershocks occurred in areas of stress increased by the mainshock.

The seismic sequence left unbroken ~30 km of the former seismic gap at the westernmost edge of the Pütürge segment. This
part of the EAF has been positively stressed by the Elazığ-Sivrice earthquake and has the potential to host a future earthquake of similar magnitude. The Elazığ-Sivrice earthquake highlights the need for accurate monitoring of the EAF, which will help to understand the fault system dynamics and mitigate the seismic hazard in Southern Turkey.

An alternating seismic activity on the NAF and the EAF faults has been observed in the earthquakes history (e.g. Agnon et al., 2006), with the two last sequences of five large earthquakes between 1866 and 1905 with 6.7<Mw<7 at the EAF,
followed by 12 large earthquakes between 1939 and 1999 along the NAF. For the EAF particularly, the generation of series



of large earthquakes, or "seismic bursts", have been described (Hubert-Ferrari et al., 2020). While there has been no activation on the NAF since 1999, the EAF experienced the Mw 6.1 Kovancilar earthquake in 2010 and now the Elazığ-Sivrice earthquake in 2020. These earthquakes may mark the returning of increased seismic activity and hazard to the EAF for the coming years.

**8 Data availability**

All data is free of charge. Strong motion data and seismic catalog were downloaded from the Disaster and Emergency Management Authority Presidential of Earthquake Department (AFAD) available at https://deprem.afad.gov.tr/ddakatalogu?lang=en. Broadband seismograms were downloaded from the permanent seismic network of Kandilli Observatory and Earthquake Research Institute (KOERI), the GFZ German Research Center for Geosciences (GEOFON) and Incorporated Research Institutions for Seismology (IRIS) Data Management Center, accessible at http://eida.gfz-potsdam.de/webdc3/. InSAR interferograms were made using Copernicus Sentinel data available at https://scihub.copernicus.eu/. Active faults were downloaded from The European Database of Seismogenic Faults (EDSF) accessible at http://diss.rm.ingv.it/share-edsf/. Last access of the AFAD strong motion data was on 25 January 2020. The other data sources were last accessed on 31 March 2020.

**9 Supplement**

The best-fitting model in point source approximation (teleseismic and regional distance) with model parameters uncertainties, moment tensor decomposition and trade-offs of the parameters are available in Figs. S7, S8, S9, S10 and S11. The best-fitting model in finite fault: strong motion waveforms fit, distributions of the parameters and their uncertainties and trade-offs between pairs of model parameters in the sequence of iterations, are available in Figs. S12, S13, S14 and S15.

**10 Author contributions**

Investigation and research was performed by all the authors. Specifically, Mohammadreza Jamalreyhani and Dr. Pınar Büyükakpınar processed seismic data and wrote original draft under the supervision of Dr. Simone Cesca, Prof. Dr. Torsten Dahm, Dr. Henriette Sudhaus, Dr. Mehdi Rezapour, and Dr. Sebastian Heimann. The ideas and research goals of this manuscript formulated by Mohammadreza Jamalreyhani and Dr. Pınar Büyükakpınar in the leadership of Dr. Simone Cesca and Prof. Dr. Torsten Dahm. Processing of the InSAR data has been done by Dr. Henriette Sudhaus and Marius Paul Isken. The Coulomb stress change calculated by Behnam Maleki Asayesh. All authors contributed significantly in many ways, e.g. by critical review and revisions.



## 11 Competing interests

The authors declare that they have no conflict of interest.

## 12 Acknowledgments

M.J. and P.B. acknowledge support by the International Training Course "Seismology and Seismic Hazard Assessment" which has been funded by the GeoForschungsZentrum Potsdam (GFZ) and the German Federal Foreign Office through the

German Humanitarian Assistance program, grant S08-60 321.50 ALL 03/19. We are very grateful to Claus Milkereit and Dorina Kroll. Furthermore, M. J. acknowledge co-funding by the University of Tehran, Iran. M.P.I. and H.S. acknowledge funding by the German Research Foundation DFG through an Emmy-Noether Young-Researcher-Grant (#276464525). We are thankful to Mehdi Nikkhoo, Ömer Emre, Sebastian Hainzl, Fatih Turhan, and Hannes Vasyura-Bathke for constructive comments on this work. Some of the maps were prepared using the Pyrocko toolbox and GMT 5 software.

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







**Figure 1: Simplified regional seismotectonic of the Anatolian block and escape tectonics. The East Anatolian Fault (EAF) and segmentation are based on Duman and Emre (2013). Seven segments on the main strand of the EAF from West to East; the Amanos, Pazarcık, Erkenek, Pütürge, Palu, Ilica and Karliova segments. All historical and instrumental seismicity during 1900-2019 and 4<=Mw<6.5 are shown as blue circles, and their occurrence within the seven different segments illustrated by red bars in the inset. Among these, the large historical earthquakes (Mw 6.5+) are given with labels. The last earthquake near to the Pütürge segment occurred in 1905, Mw 6.8. The cross-section shows the depth of events with 4<Mw<6.5 during 1900-2019 in the EAF. Black star shows the epicenter of the Elazığ-Sivrice earthquake. Red lines indicate active faults in the region (Basili et al., 2013).**

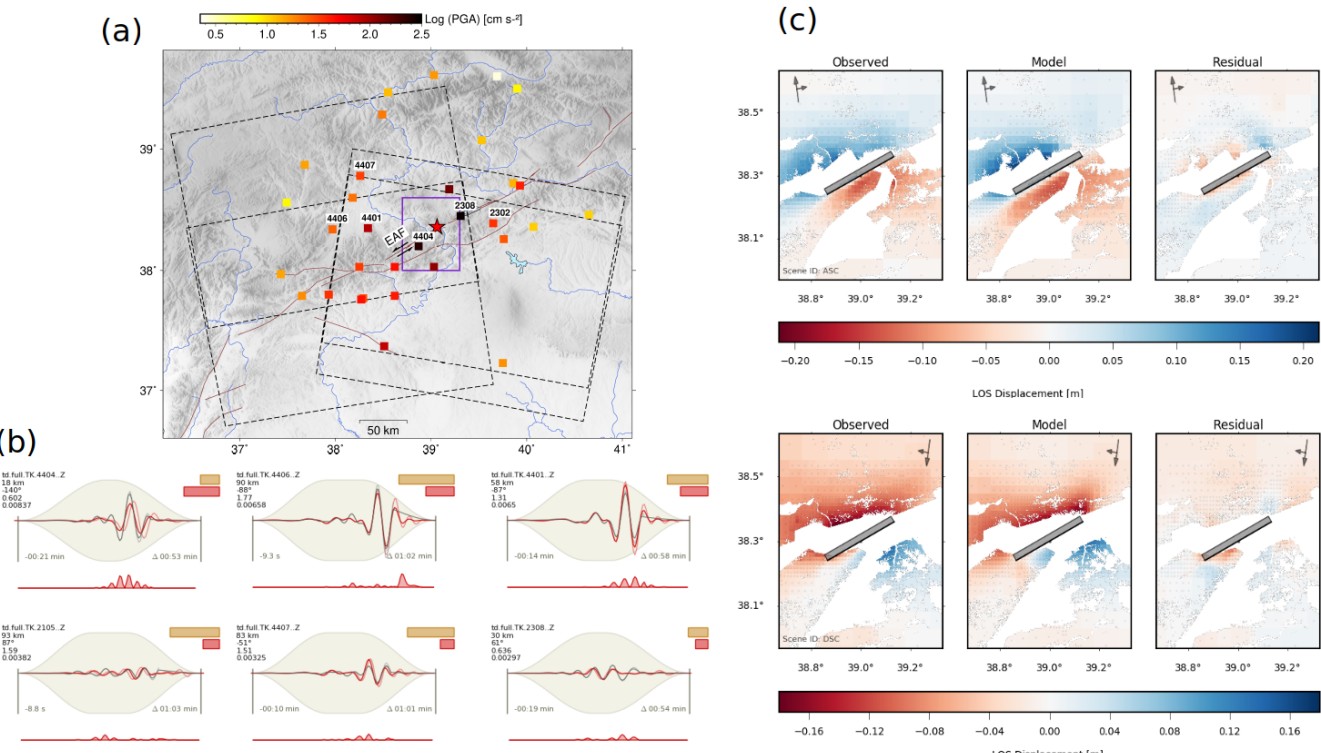

**Figure 2: Station and data coverage in the epicentral area with examples of data and forward modeling. a) The squares show strong motion stations colored according to peak PGA values. Stations with codes are those used in the joint inversion. The dashed black boxes indicate the spatial extent of used Sentinel-1 imagery from both ascending and descending orbits. The red star shows the epicenter of the mainshock. Red lines indicate active fault maps in the region (Basili et al., 2013). The purple box shows the spatial coverage in the panel c. b) Strong motions modeling: The best-fitting model in the Z component of 6 near-field strong motion stations; observed trace (dark gray) and synthetic trace (red). Information in the waveforms fit (left side, from top to bottom) gives station name with the component, distance to the source, station azimuth, weight, misfit and starting time of the waveform (relative to the origin time). c) InSAR modeling: Subsampled surface displacements as observed, modeled and with the data residual. The grey filled box shows the surface projection of the modeled source, with the thick-lined edge marking the upper fault edge.**







**Figure 3: Spatiotemporal evolution of the 2020 Mw 6.77 Elazığ-Sivrice earthquake sequence (black stars always denote the mainshock, while brown solid and hollow purple circles show fore- and after-shocks respectively). a) spatial distribution of seismicity at the Pütürge segment, located between the Hazar Lake and the Yarpuzlu bend, showing the path of Firat River (blue line), which crosses the EAF with an 11 km left-lateral offset since the Pliocene (Duman and Emre, 2013), main faults (red lines, after Basili et al., 2013), epicentral locations of fore- and after-shocks (circles, Ml 1+ and azimuthal gap less than 120°), focal mechanisms of the mainshock, 2 foreshocks and 16 aftershocks (focal spheres, color scale according to centroid depths). Black squares denote locations of the closest strong motion stations with their code. b) Depth cross-section of the events larger than Ml 4, showing the rupture area (red rectangle) and direction of rupture propagation, as resolved in this study (the rupture propagated almost unilaterally toward WSW); two large foreshocks (brown dots) are located close to the mainshock nucleation point. c1, c2, c3) Depth cross-sections along profiles AB, CD and EF, respectively (dip and width of all cross-sections are 90° and 20 km, respectively), showing the focal mechanisms of largest events (cross section projection). d) Temporal evolution of the aftershocks (Ml 1+ and azimuthal gap < 120°) versus longitude; the upper histogram shows the longitude versus the logarithm of the number of events, the lower number of aftershocks activity in the central region of ~25 km length, which matches the location and extent of the mainshock rupture area (light yellow region). e) Temporal evolution of the foreshocks (same style as panel d); two foreshock clusters located close to the mainshock nucleation point (4 April 2019 Mw 5.2 and 27 December 2019 Mw 4.9).**





**Figure 4: Coulomb stress changes due to the 2020 Elazığ-Sivrice earthquake: a) on the surrounding faults with the same orientation as the causative fault and depth of 7.0 km. b) on to the stress field optimally oriented faults and distribution of aftershocks. Aftershocks are shown with green dots. The black stars, and black lines show the epicenter of the mainshock and active faults, respectively.**





**Table 1: Source mechanism solutions of the 24 January 2020 Mw 6.77 Elazığ-Sivrice earthquake with parameters uncertainties (68% confidence intervals). Moment tensor (M. T.), Finite Fault (F. F.). GCMT: Global Centroid Moment Tensor. GEOFON: GFZ German Research Center for Geosciences. AFAD: Disaster and Emergency Management Authority Presidential of Earthquake Department. KOERI: Kandilli Observatory and Earthquake Research Institute. USGS: U.S. Geological Survey-National Earthquake Information Centre.**

680

| Reference | Method | Data | Latitude° | Longitude° | Strike° | Dip° | Rake° | Depth (km) | Length (km) | Width (km) | Slip (m) | Duration (s) | Velocity (m s⁻¹) | Moment (N.m) | $M_w$ |
|---|---|---|---|---|---|---|---|---|---|---|---|---|---|---|---|
| GCMT | M. T. | Teleseismic | 38.3 | 39.0 | 246 | 67 | -9 | 12 | - | - | - | 11.8 | - | 1.77E+019 | 6.8 |
| GEOFON | M. T. | Teleseismic | 38.35 | 39.19 | 245 | 81 | -21 | 20 | - | - | - | - | - | 1.80E+019 | 6.8 |
| AFAD | M. T. | Regional | 38.3593 | 39.063 | 248 | 76 | 1 | 15.1 | - | - | - | - | - | 3.05E+019 | 6.8 |
| KOERI | M. T. | Regional | 38.52 | 39.29 | 248 | 87 | -4 | 10 | - | - | - | - | - | 1.29E+019 | 6.7 |
| USGS | M. T. | Teleseismic | 38.39 | 39.09 | 245 | 80 | -12 | 21 | - | - | - | 11.5 | - | 1.39E+019 | 6.7 |
| USGS | F. F. | Teleseismic | 38.3 | 39.1 | 246 | 67 | -12 | 10 | 40 | 10 | 1.7 | 20 | | 1.80E+019 | 6.8 |
| This Study | M. T. | Teleseismic | 38.25±0.03 | 38.90±0.08 | 247±6 | 70±6 | -5±6 | 8±2 | - | - | - | 27±1 | - | 1.37E+019 | 6.7±0.2 |
| This Study | M. T. | Regional and Teleseismic | 38.37±0.02 | 39.01±0.06 | 246±3 | 70±9 | -16±9 | 5±2 | - | - | - | 28±1 | - | 1.79E+019 | 6.7±0.1 |
| This Study | F. F. | InSAR and Strong Motion | 38.32±0.01 | 39.0±0.02 | 242±1 | 74±2 | 0±2 | 5.5±1 | 26±5 | 9±1 | 1.6±0.2 | - | 2100±130 | 1.3±0.4E+019 | 6.6±0.1 |