# Peer review of "Data overview and local velocity models"

_Solid Earth, 2020_

## Referee Comment (RC1) · Anonymous Referee #1 · 23 Jun 2020

The study presents a detailed analysis and interpretation of the 2020 January 24 Mw 6.7 The ElazĜħ-Sivrice earthquake. The study presents a full geodetic and seismological analysis of the source rupture and the peripheral seismicity including foreshocks and aftershocks.

Coulomb stress and seismicity pattern – Interestingly, the south part of the fault has no aftershocks. This seismic quiescence is puzzling even more due to the prediction of the Coulomb stress analysis and the symmetry in the InSar data from both sides of the fault supporting that surface deformation took place at the south. Do you have

explanation for that behavior?

Specific comments: • L. 24 – Please use quantitative rather than "small" for the described foreshock cluster • L. 29 – Please explain how the statement for shallow locking depth corresponds with the seismicity range presented in Fig. 1 (0 – 30 km) • L. 118 – Please explain the usage of strong motion sensors to capture low frequency signal • L.131 – Did the earthquake rupture to the surface? This is not clear • L. 206 – It is not clear to me how did you conclude that the mainshock nucleated from the topper part of the fault plane from Fig. 3b. Please elaborate. • Fig. S5 – This is a very nice presentation of a unilateral rupture. I think it should be included in the main text. Please also consider presenting the seismic traces with azimuth to support the apparent duration measurement.

---

## Referee Comment (RC2) · Anonymous Referee #2 · 29 Jun 2020

The present study examines the 24 January 2020 Elazig-Sivrice earthquake by jointly trying to model quasi co-seismic static surface displacements from InSAR and high-frequency co-seismic data from seismological networks at local, regional and teleseismic distances to retrieve source parameters of the mainshock. Furthermore, the authors claim that they estimated moment tensor for 18 fore-/after-shocks with Mw $\geq$ 4.3 based on the modelling of the regional broadband data. The authors declare and highlight that the mainshock partially ruptured a seismic gap. Although the current work examines an important event occurred recently in the region, I do have problems with

the present manuscript particularly because of some significant principle seismological issues which I have attempted to clarify and to explain those point effectively along the lines detailed below.

The organization of the manuscript and presentation of the data and results need significant improvement with major revisions, clarification and organization to focus it on its most interesting topic, the one announced in the title. Therefore, I, alas, find the current status of the paper very poor and it is NOT scholarly written in many ways.

Briefly, I, first of all, find the present work very weak mainly because of an insufficient of visual materials to support the authors' arguments such as unilateral slip characteristics or so. Secondly, discussion section lacks a decent organization and, thus, it needs more comprehensive assessment and interpretations of the results. Unfortunately, at most part of the text we see superficially obscured questions. For instance, the link between InSAR based models and interpretation on fault segmentation is barely discussed within only two lines of sentences. Furthermore, I found some part of the discussion in which various scenarios are compared based on the aftershock distribution is very misleading due to the inappropriate resolution capability of the aftershock distribution data obtained from the AFAD PDE catalogues...

I feel that the manuscript written in a hurry is rather unfocussed and could also clearly benefit from careful editing by a native speaker as the written English needs some brushing up. Overall, this manuscript must have substantial changes in the present form (e.g., improvements in manuscript title, abstract, introduction, organization, layout, re-writing, figures, references, discussion and conclusions etc.) before further consideration for the EGU-Solid Earth as a brand-new submission.

Thus, it is NOT suitable and NOT acceptable for a publication at the Solid Earth journal as it is.

I recommend REJECTION and resubmission to SE and/or to any other journals.

* Please see attached supplementary PDF file containing my anonymous referee report *

Please also note the supplement to this comment:
https://se.copernicus.org/preprints/se-2020-55/se-2020-55-RC2-supplement.pdf

**Supplement:**

*"Seismicity related to the eastern sector of Anatolian escape tectonic:
the example of the 24 January 2020 Mw 6.77 Elazig-Sivrice earthquake"*
* * *
*General Comments:*
* * *
The present study examines the 24 January 2020 Elazig-Sivrice earthquake by jointly trying to model quasi co-seismic static surface displacements from InSAR and high-frequency co-seismic data from seismological networks at local, regional and teleseismic distances to retrieve source parameters of the mainshock. Furthermore, the authors claim that they estimated moment tensor for 18 fore-/after-shocks with Mw ≥ 4.3 based on the modelling of the regional broadband data. The authors declare and highlight that the mainshock partially ruptured a seismic gap. Although the current work examines an important event occurred recently in the region, I do have problems with the present manuscript particularly because of some significant principle seismological issues which I have attempted to clarify and to explain those point effectively along the lines detailed below.

The organization of the manuscript and presentation of the data and results need significant improvement with major revisions, clarification and organization to focus it on its most interesting topic, the one announced in the title. Therefore, I, alas, find the current status of the paper very poor and it is NOT scholarly written in many ways.

Briefly, I, first of all, find the present work very weak mainly because of an insufficient of visual materials to support the authors' arguments such as unilateral slip characteristics or so. Secondly, discussion section lacks a decent organization and, thus, it needs more comprehensive assessment and interpretations of the results. Unfortunately, at most part of the text we see superficially obscured questions. For instance, the link between InSAR based models and interpretation on fault segmentation is barely discussed within only two lines of sentences. Furthermore, I found some part of the discussion in which various scenarios are compared based on the aftershock distribution is very misleading due to the inappropriate resolution capability of the aftershock distribution data obtained from the AFAD PDE catalogues…

I feel that the manuscript written in a hurry is rather unfocussed and could also clearly benefit from careful editing by a native speaker as the written English needs some brushing up. Overall, this manuscript must have substantial changes in the present form (e.g., improvements in manuscript title, abstract, introduction, organization, layout, re-writing, figures, refences, discussion and conclusions etc.) before further consideration for the *EGU-Solid Earth* as a brand-new submission. Thus, it is NOT suitable and NOT acceptable for a publication at the *Solid Earth* journal as it is.

I recommend **REJECTION** and resubmission to SE and/or to any other journals.

*Scientific Rationale:*

*Abstract:*

The authors state that "*2020 Elazig-Sivrice mainshock shows that the earthquake, with a magnitude Mw 6.77, ruptured at shallow depth (5±2 km) with a left-lateral strike-slip focal mechanism, with a dip angle of 74°±2° and a causative fault plane strike of 242°±1°, which is compatible with the orientation of the EAF at the centroid location*". Why did they accept this solution of the 2020 Elazig earthquake? However, in text, they reported three diverse focal mechanism parameters from inversions with different data set (see the Table 1; strike/dip/rake/depth/Mo/Mw). But, in Abstract, they mixed all parameters. For example, they selected strike and dip angles of FF (InSAR and Strong Motion), but they selected focal depth and Mw from MT (with Regional and Teleseismic data). So, what is the scientific motivation for these choices in describing the focal mechanism parameters of the 2020 Elazig earthquake? Therefore, Abstract needs serious corrections in many ways.

| This Study | M. T. | Teleseismic | 38.25±0.03 | 38.90±0.08 | 247±6 | 70±6 | -5±6 | 8±2 | - | - | - | 27±1 | - | 1.37E+019 | 6.7±0.2 |
| This Study | M. T. | Regional and Teleseismic | 38.37±0.02 | 39.01±0.06 | 246±3 | 70±9 | -16±9 | 5±2 | - | - | - | 28±1 | - | 1.79E+019 | 6.7±0.1 |
| This Study | F. F. | InSAR and Strong Motion | 38.32±0.01 | 39.0±0.02 | 242±1 | 74±2 | 0±2 | 5.5±1 | 26±5 | 9±1 | 1.6±0.2 | - | 2100±130 | 1.3±0.4E+019 | 6.6±0.1 |

*Methods and Results: general comments*

In general, introduction, method and results sections (e.g., mainshock, seismic sequence, Coulomb failure stress change analysis) are NOT clearly presented and NOT well-written. For example;

**A.** The details of each methodology used in this work are not adequately explained here. The authors only described what they did in this study. No detailed information about the processing steps are provided on the selected algorithms (e.g., Grond, SNAP, Kite, Coulomb 3.3 etc.) There are too many unexplained sections about them making it difficult for the readers. They only directed the readers to the *Pyrocko* webpage for getting the information on each algorithm. However, I think it would be appropriate to present additional details clarifying each method and work devoted, since it is an important part of the article. Furthermore, the authors did not present their results in appropriate ways. The results section contains missing and incomprehensible parts. For example, on page 6, the authors reported that "*We performed a moment tensor inversion for 18 earthquakes (2 foreshocks and 16 aftershocks) with Ml ≥ 4.3. For this purpose, we proceed as for the point source inversion. However, due to the weaker magnitude, we rely on only regional broadband data of the KOERI network (Fig. S3)*". But they did not provide any BWIDC or surface-wave point-source MT results in Fig. S3 which only presents wrapped and unwrapped interferograms spanning the co-seismic of the Elazig earthquake for both ascending (up) and descending (down) directions. The authors should be very careful in numbering and referring to figures…

**B.** Also, the authors provided time domain waveform fits for some selected traces for the best model at teleseismic (P- and SH- waves: tp.p and td.s) and regional (Rayleigh and Love waves: rd.rayleigh and td.love) distances. But they did not summarize the obtained/preferred source parameters (e.g., source mechanism models) from these waveform fits.

Hence, I think the results are generally not displayed properly and satisfactorily presented in Fig. S7. There are too many points to criticise and to question further such as:

1. Did the authors also cross-check P-wave first motion polarities recorded at near-field stations? Did they examine if the P-wave polarities are compatible with the nodal planes or not?

2. The authors compared their focal mechanism solutions with those source parameters reported by other agencies (GCMT, GEOFON, AFAD etc., see Table S1). What are the main differences between these solutions? Did they check the effects of variations in each source parameters on waveform fits? Why do they suggest that their parameters are more reliable than the other solutions? Authors should verify and stipulate additional figures/plots, maps in order to convince readers that their results significantly appropriate and better than the others.

3. How did the authors calculate the uncertainties of each source parameters? Which method was used to determine the amount of uncertainties in the source parameters? There is crucial need to clarify these points (see page 4 lines 110-115).

4. Why did the authors select the frequency band of 0.08-0.20 Hz in the finite source optimization/inversion with near field data? Similarly, the authors modelled entire waveforms and amplitude spectra in the frequency band ranging 0.02-0.05 Hz. How and why did they select these frequencies?

5. Is there any slip distribution/rupture propagation model with the amount of displacements on the fault plane for which the authors favour? How did they estimate the fault length and fault width for this earthquake? They barely provided some waveform fits that are not clearly recognised (see comment E on *Discussions and Conclusions*).

6. They also did not evidently explain that which source parameters (Table 1 or else?) were used in Coulomb stress change analysis. They only mentioned that a homogeneous elastic half-space Earth model and *the causative fault* of the Elazig-Sivrice earthquake are considered as a rectangular dislocation (26 km long 9 km wide with mean slip equal to 1.8 m) in Coulomb stress modelling. What is the **causative fault plane** of this earthquake? The authors should distinctly summarize each earthquake source parameters resolved (e.g., strike/dip/rake angles/seismic moment/depth etc) that they used in Coulomb stress analysis. Furthermore, the authors should explain how they calculated the mean slip of 1.8 meters given in this section too (see Page 6 line 190).

7. How did the authors estimate the rupture duration and rupture velocity? (see page 5 line 135).

C. Nonetheless, my biggest concern is primarily on how the authors describe the rupture propagation and its time evolution in the current manuscript. Neither in the text nor in the supplementary material they present convincing material documenting the signature of unilateral propagation that the authors generously claimed. I would expect to see spatio-temporal co-seismic slip behaviour following the inverse modelling of such data set, if any. However, this is a very critical detail regarding the physics of time evolution of this earthquake, and there is NO

evidence in this current work to clarify and/or to debate on these diversities of observations and interpretations.

In fact, if there is a proposed model of co-seismic slip distribution based on the inversion of InSAR data set, I have not seen any relevant model result and I am wondering why authors avoided to share these details, if any. The absence of segmentation is only mentioned very briefly referring to the InSAR data modelling in Fig. 2c (Page# 8, Lines#: 245-247, see: "*Some systematic residuals in the near-fault InSAR results of the finite slip modelling (Fig. 2) may point to a slight segmentation, but the overall good data fit in the single-segment finite fault modelling suggests that segmentation is not a first-order feature.*"). But unfortunately, I cannot see any clear elaboration from the interpretation of Fig. 2c. Even it is exceedingly unclear to what these two different InSAR modelling results belong which specific data subset. At this stage answer to this issue is very critical because the reader can have tough times in understanding the link between the rupture process and InSAR data with only available information presented in the current form of the manuscript. It would be nice to see snap-shots and a movie of time evolution on the preferred model of co-seismic slip distribution using InSAR data-set.

**D.** Another surprising issue for me is that why authors did not consider a direct strong motion data analysis obtained from a fairly dense station distribution though this is also questionable issue (?) as the Turkish Government AFAD authority **officially** released the data set on the **16 June 2020**, and not before! Hence, I am NOT contented how the authors attained this data-set which was not available on the days of 24-25 January 2020 and afterwards until the 16th June 2020.

**Thus, I am quite curious how the authors obtained these unreleased strong-motion data which should be clarified and confirmed in writing from the Turkish government authorities. Otherwise, this does NOT grant an equal opportunity on *Data Availability* for international scientists to conduct a research on the current and other relevant earthquakes in the region for global and/or regional mutual interests. Therefore, I consider this current work being NOT an objective piece of scientific conduct, and it is quite unfair to the others interested to study these events further.**

Furthermore, under these conditions, one would investigate the time variation of the pseudo displacement that could be easily extracted from these near-field recordings in relation to the ground motion using these stations located at different azimuths with respect to the mainshock. This, therefore, would give a direct and less-biased information on likely different episodes of the propagation. The authors should revisit these issues to elucidate further. And, it would be healthier to see snap-shots and a movie of time evolution on the preferred model of co-seismic slip distribution based solely on strong-motion data-set.

**E.** Page# 8, Lines#: 243-245, "*Scenario 3 is unlikely, because the depths estimated by earthquake relocation and aftershock centroid MT inversion do not change significantly along the fault. Both scenarios, 1 and 2 are plausible and in contradiction with the observed data*"

I do not think that the AFAD's routine PDE locations have very high resolution that can enable us a precise aftershock distribution to comment on further. It is pretty clear that they are not quite reliable to make such firm conclusions. The biggest problem with their relocated earthquakes

stems from the very irrelevant type of 1-D initial seismic velocity structure model used in their localization procedure. Thus, the best option would be to perform relocations based on conventional relative techniques since they can provide much precise values as they do not depend on presumably uncertain knowledge of seismic velocity structure. Relative locations, for instance, HypoDD will better work in keeping track of the spatio-temporal behaviour of the seismicity much consistent. Precise relocation is highly achievable via phase reading data set that is publicly accessible dataset from the AFAD and other regional archives of KOERI or so (*see:* Waldhauser F. and W.L. Ellsworth, 2000. *A double-difference earthquake location algorithm: Method and application to the northern Hayward fault, Bull. Seism. Soc. Am., 90, 1353-1368, 2000;* Waldhauser, F., 2001. *HypoDD: A computer program to compute double-difference earthquake locations, USGS Open File Rep., 01-113, 2001*).

On the other hand, foreshocks can be often described as small event activities in a close proximity to the hypocenter of the mainshock. They may have an essential role in understanding the physics and initiation process of an upcoming event. Two main models have been, so far, proposed to explain the link between foreshocks and the main rupture. These are pre-slip (Ellsworth and Beroza, 1995) and cascade models (e.g. Fukao and Furumoto, 1985). In order to efficiently evaluate whether a series of events can be regarded as the foreshock activity and the proper mechanism involving the type of physical process affecting on the fault plane requires a tedious and critical investigation on extremely precise event localization, spectral analyses for their high resolution source characteristics (e.g. source radius, released energy, etc.), or the amount of stress change they caused. Although, the present work refers to possible foreshock activities at few places in the text, it is hard to see reasonable arguments if these activities can be interpreted within the concept of foreshock classification. There are almost no detailed efforts performed in the present work to elucidate this issue.

*Figures:*

Most of the figures are NOT well prepared for a clear publication quality. They are rather busy, and there must be a way to make them look easier to read. The fonts used on the maps and seismograms (*see* Fig. 2a, b, c; Fig. S7, S11, S13, S14, S15) are too small, hence it is hard to read. There are also many missing sections and references as such in the figure captions. The authors should cautiously arrange/plot the figures and re-write the figure captions without leaving any open questions as far as copyright issues are concerned.

**Fig. 1 and Inset:**

The *Inset* is very confusing as it does not present right geometry of the NAFZ in the Sea of Marmara, nor the extensional features in the western Turkey and the Aegean Sea. There is immense amount of high-quality papers, graduate thesis and extensive geophysical and geological experiments conducted in the Sea of Marmara and the Aegean regions, but I do NOT see them noted in discussions or cited in references leaving many open questions. What are the sources of historical and instrumental earthquake data and GPS velocity vectors plotted in Fig. 1? Any references to add?

**Figs. 2a, b,c; S2 and S4:**

In Fig. 2a, the axis information (i.e., coordinates of the location map) is too small and not readable at all. The readers cannot easily recognise which stations are in the rupture direction and which are in the opposite direction to the rupture zone (see Fig. 2a) as the names/codes of stations are not properly presented on a focal sphere (i.e.: azimuthal coverage), and/or on the map view (see Fig. S2). In addition, the number of seismic stations identified on the map (see Fig. 2a) is much more than the number of waveforms modelled. Explain the main reasons why the other stations are not modelled in the joint inversion. How did you select the modelled stations? What are the criteria? The authors should provide a summary Table specifically to elucidate Fig. 2a, and it can be posted to the supplementary.

Fig. 2b is not helping readers to identify the waveform fits, and details of the methods and software should be briefly summarized in the Supplementary On-Line material to guide those who are not familiar with this code. Besides, the choice of coloured lines and envelope is not helpful, therefore there is a need for further upgrade here. Fig. 2c requires some more details regarding the difference between two different solution. What is the major observation, interpretation and discrepancies between top and bottom InSAR maps in Fig. 2c?

**Figs. 3. and S1:**

The entire Fig. 3 is NOT acceptable at all. It should be seriously revised. The aftershock distribution in this figure does not make any sense due to too much uncertainty inherently existing in the epicentre/hypocentre location data taken from the AFAD PDE data catalogues. I strongly suggest that the imperative data must be revisited by using relative relocations and/or any other conventional methods. The authors should add a proper reference of the velocity model to the caption of Fig. S1, and also give right reference to the AFAD for earthquake locations plotted in Fig. 3. The discussion based on these maps and figures are irrelevant, and they should be removed in text as it does NOT reflect the ground truth until authors improve them with the accurate relocation techniques. How the rupture area (red rectangle) and rupture direction in Fig. 3b are defined, likewise light-yellow coloured region in Fig. 3d, e? Is it simply based on inadequate seismicity map of the AFAD and/or KOERI or else, if any?

**Fig. 4:**

As long as the aftershock distribution data is updated, presenting the vertical extent of the stress change along the fault plane can be physically more meaningful as well. The authors should add a proper reference for active faults plotted in the epicentral area. This figure should be presented in the supplementary along with other Coulomb stress change map in Fig. S6 with additional brief information on CST and up-to-date worthy referencing.

**Table 1:**

Source mechanism solutions are summarized, but I am still not quite sure how the authors claim that the magnitude of mainshock is Mw 6.77. Where is it taken from? How about the errors? How did you calculate them? The authors should provide parameters of their own results in a separate Table.

There is NO introductory information regarding these material as the authors should add a cover page describing individual plots. It looks quite clumsy as a grab bag in its present form.

**Fig. S1:**

What is the source for the 1-D radial velocity models used for calculating near-field Green's functions? Any references? Is Acarel et al. (1996) right one to refer to? According to my recollection, this region is characterized by relatively thick crust compared to the rest of Turkey and Moho depth taken about 31-32 km in this model could be misleading. I highly recommend authors to check recent papers dealing with Anatolian crust based on ambient noise tomography (e.g., Delph et al., 2015), Pn tomography tomography (e.g., Mutlu and Karabulut, et al., 2011), receiver functions (e.g., Vanacore et al., 2013; Karabulut et al., 2019) in order to obtain a reliable 1-D velocity model, among many other tomography studies in 1-D and even furthermore in 3-D FWI studies.

**Fig. S2:**

Here, the distribution of teleseismic stations are plotted without station codes. Thus, it is making even more difficult for readers to identify waveform fits. Besides, there is a huge azimuthal gap in the North and North-East quadrants spanning from Greenland to Kamchatka peninsula? This is quite important especially when the authors speak about directivity of the main fault based on their observation at other data-set they claim that they have along NE-SW striking geometry. How can we see these propagation effects (i.e., doppler-shifting) in waveforms if we do not have stations at these azimuths? And, also, in SW azimuths. The authors should provide complete list of stations for which broadband *P*- waveforms obtained along with complete catalogue information in a Table. Additionally, arcs of latitudes and longitudes should be plotted at each 15º arc-distances or so in order to help readers for an orientation of the nodal planes.

**Fig. S3:**

The more detailed information should be provided in figure caption for wrapped and unwrapped interferograms of the InSAR data presented in Fig. S3.

**Fig. S4:**

This is a confusing map… Firstly, traces and/or outline of main fault zones in Turkey, in the Aegean and nearby countries are NOT precise and misleading as presented in inset of Fig.1. The authors should be very careful in cross-referencing others' data without proper knowledge. This figure ought to be replaced with right one as there are many experts around to give help. Otherwise, copy and paste fashion can be very damaging one, and it looks as if data-base of the main faults of Basili et. al. (2013) is NOT the right resources to make use of it…? Hence, remove this reference and use a decent relevant one on the active faults of Turkey and surroundings. Secondly, what are the sources and which networks of coloured triangles refer to? Green? and Red? triangles stand for which network? Are there misinformation here?

How about AFAD broadband stations as the authors were able to get some of their other type of dataset? Why those local and regional broadband seismic stations of AFAD are not used in joint inversions? Any clarification and/or explanation?

**Fig. S5:**

It would be better to mark the units of the colour scale-bar (s stand for seconds?) given next to the plots?? Above all I find this figure not a helpful one.

**Fig. S6:**

This figure should be presented along with Fig. 3 and with additional brief information on CST and up-to-date new referencing. However, it is not informative without inclusion of the mechanisms of entire clusters with proper seismicity. Otherwise, this can be removed from the manuscript.

**Fig. S7:**

This figure presents time domain waveform fits of selected P- and SH- body waves, and regional surface waves for the mainshock. However, the distribution of teleseismic stations are plotted without station codes in Fig. S2 which makes it difficult to analyse these closely. Thus, it is making even more difficult for readers to identify waveform fits. Besides, there is a huge azimuthal gap in the North and North-East quadrants spanning from Greenland to Kamchatka peninsula? It is NOT proper to present automated figure generations in the Grond software toolbox of Heimann et al. (2018). The authors should refine these graphics to be more relevant for the readers to orient themselves. Furthermore, brief introductory explanations should be provided in order to summarize the main features of the Grond software toolbox. Specifically, filtering is dangerous, and should be clarified properly reasons why. Otherwise, it looks like an output of the computing in a black-box fashion.

**Figs. S8-S13:**

The authors should add colour bars for the misfit values. Also, it would be great to contribute some explanatory information regarding the optimization procedure used throughout in this article. For instance, I am not even sure what type of data you are displaying as misfits? Is this the total misfit obtained from the contribution of different data sets (strong motion, teleseismic, geodetic, etc.)? The detailed clarification is needed as this issue is rather critical.

**Fig. S8:**

Sequence plots of distribution and uncertainties of some parameters are a bit confusing one, and it does not help much with 68% confidence intervals.

**Fig. S9:**

Bootstrap misfit of the optimization is also too technical and does not help the readers much. Therefore, this figure should be removed.

**Fig. S10:**

Yes, I agree that MT decomposition is not well presented, and requires further analyses. Therefore, this figure should be removed.

**Fig. S11:**

Source parameter's scatter plots are not easily readable and does not help the readers much. Therefore, this figure should be removed especially when considered the azimuthal gaps of the broad-band stations used.

**Fig. S12:**

It refers to time domain waveform fits for strong motion data. Again, I do have reservations on this data-set and how the authors were able to get an access these data before the official release date of the **16 June 2020**. So where is the doppler effects and directivity on strong-motion data? Also, it is NOT proper to present automated figure generations in the Grond software toolbox of Heimann et al. (2018). The authors should refine these graphics to be more relevant for the readers to orient themselves. Furthermore, brief introductory explanations should also be provided in order to summarize the main features of the Grond software toolbox. Specifically, filtering is dangerous, and should be clarified properly reasons why. Otherwise, it looks like an output of the computing in a black-box fashion.

**Fig. S13:**

Finite Fault model plots of distribution and uncertainties of some source parameters of FF are a bit confusing one, and this figure does not help much to convince the reader especially when considered the azimuthal gaps of the broad-band stations used. Therefore, this figure should be removed.

**Fig. S14:**

Bootstrap misfit of the optimization for the FF model is also too technical and does not help the readers much. Therefore, this figure should be removed.

**Fig. S15:**

Source parameter's scatter plots for the FF model are not easily readable and does not help the readers much. Therefore, this figure should be removed especially when considered the azimuthal gaps of the broad-band stations used.

**Table S1:**

Moment tensor inversion results of the foreshocks and aftershocks are summarized in Table S1. However, I would like to see individual plots of complete waveform-fits of each earthquake spanning from 4 April 2009? right date? Otherwise, from 27 December 2019 to 19 March 2020. Are they regional moment tensor (RMT) results or else?

How reliable are these mechanisms? How about the error bars in the earthquake mechanisms of both nodal planes? The waveform modelling for earthquake and tsunami source studies is a tedious profession and it takes longer time and careful consideration. Thus, I advise authors to be very careful at this kind of studies.

*References:*
* * *
The authors are using selective limited publications to cite, and some of them are irrelevant ones. Besides, there are very valuable SCI journal papers and Special Issues of WoS Journals as well as the known established society's special publication Books to cover specific questions on neotectonics, seismotectonics and geodynamic evolution of the eastern Mediterranean Sea region and Anatolia ranging from seismology, geodesy, geochemistry to 1-D/3-D teleseismic and local earthquake tomographic studies to orient the authors and the readers. The authors need to invest some further reading sessions on the above topics regarding the Eastern Mediterranean Sea region.

1. Acarel et al. (1996) paper is an irrelevant and poor one and not an objective good quality paper to cite as there are major misleading information included. Are you aware of them? Therefore, the adapted local crustal model is not valid and not reliable one to rely on further.

2. I do not see quite relevance of the below articles besides being case studies.

   I advise removal of one of the below articles?

   -Asayesh, B. M., Hamzeloo, H. and Zafarani, H.: Coulomb stress changes due to main earthquakes in Southeast Iran during 1981 to 2011. J Seismol 23, 135–150, https://doi.org/10.1007/s10950-018-9797-y, 2019.
   -Asayesh, B. M., Zafarani, H., and Tatar, M.: Coulomb stress changes and secondary stress triggering during the 2003 (Mw 6.6) Bam (Iran) earthquake. Tectonophysics, 775, 228304, https://doi.org/10.1016/j.tecto.2019.228304, 2020.

3. The statement in *Lines of 60-64* is not true as Bulut et al. (2012) was not the first to report.

   "*Bulut et al. (2012) characterize the EAF as a left-lateral strike-slip system, involving NE-SW and EW oriented segments which run parallel to the segmented trend of the main fault. Besides the dominant strike-slip mechanisms, Bulut et al. (2012) found evidence for additional thrust faulting on EW trending structures and normal faulting on NS trending secondary faults.*"

   I advise authors carefully to read scholarly written papers on the Anatolian seismotectonics and geodynamics studies in order not to reach such strong conclusions. There are many sentences like these throughout the manuscript as authors are misusing cross-referencing, and therefore not giving the right credit who deserves much in the first place. I repeat here again that the authors need to invest some further reading sessions on the above topics.

4. It looks as if data-base of the main faults of Basili et. al. (2013) is NOT the right resources to make use of it? Subsequently, remove this reference and use a decent relevant one on the active faults of Turkey and surroundings. The neotectonics features of the Anatolia is well studied and established and is widely known. So why to refer to an incomplete data-base?

Basili, R., Kastelic, V., Demircioglu, M. B., Garcia Moreno D., et al.: The European Database of Seismogenic Faults (EDSF) compiled in the framework of the Project SHARE. http://diss.rm.ingv.it/share-edsf/, doi: 10.6092/INGV.ITSHARE- EDSF, 2013.

5. Line 131, the authors report that "*Some surface cracks, rockfalls, landslides, and liquefaction were reported (Lekkas et al., 2020)*".

   Lekkas et al. (2020) did not execute field excursions after the mainshock in the area to map and to report such observations. This is not right referencing, and proves another example of wrong usage of cross-referencing! Check Turkish official report of MTA (2020) at the right web page. Otherwise, one can easily form a paper simply navigating at the virtual space to get information in a copy and paste fashion. This is a serious issue and can be considered as a misconduct as decent piece of scholarly science requires sensitive and careful analyses ever.

   **MTA. (2020).** Preliminary field and evaluation report on 24 January 2020 Sivrice (Elazıg) *Mw 6.8* Earthquake, General Directorate of Mineral Research and Explorations of Turkey (MTA), Ministry of Energy and Natural Resources, Ankara, 48 pages (https:// https://www.mta.gov.tr/).

6. The authors should also consider large aftershocks observed striking NE-SW along the EAFZ before jumping on wrong conclusions with those of Nissen et al. (2019). Thus, what is the direct relevance of the below article in the current study? I would have written a serious comment on the below article, but I do not have much time to invest on this adventure.

   Nissen, E., Ghods, A., Karas.zen, A., Elliott, J. R., Barnhart, W. D., Bergman, E. A., Hayes, G. P., Jamal-Reyhani, M., and et al.: The 12 November 2017 M w 7.3 Ezgeleh-Sarpolzahab (Iran) earthquake and active tectonics of the Lurestan arc. Journal of Geophysical Research: Solid Earth, 124. https://doi.org/10.1029/2018JB016221, 2019.

*Data Availability:*

I am quite curious how the authors obtained the unreleased AFAD's strong-motion data which should be clarified and confirmed in writing from the Turkish government authorities. Otherwise, this does NOT grant an equal opportunity on *Data Availability* for international scientists to conduct a research on the current and other relevant earthquakes in the region for global and/or regional mutual interests. Therefore, I consider this current work being NOT an objective piece of scientific conduct, and it is quite unfair to the others interested to study these events further. Similarly, why did the authors NOT use any waveforms from the local and regional broadband stations operated by the AFAD?

*Software Availability:*

Some of the tools are available for the broad scientific studies, but the details and decent expertise are rather limited. This issue should be enhanced in the text with right referencing, and note as SOM.

I feel that the manuscript is rather unfocussed and could also clearly benefit from careful editing by native speaker as the written English needs some brushing up. I can point out several places where this needs to be done below, but certainly not every occurrence.

Line 206 reads "*The mainshock started to nucleate from the topper part of the fault plane (Fig. 3b)*".

What does "topper" mean? Any good grammar? British/American English or a slang word invented?

*Discussions and Conclusions:*

In addition, I would like to hear the authors' overall comments on the following submitted and accepted articles that I have recently acquired on the dedicated web pages.

A.  I have just noticed the following accepted article on the 2020 Elazig earthquake, which is on line since 29 March 2020 under URL (https://www.essoar.org/doi/10.1002/essoar.10502613.1), and I wonder how and why authors did not note/comment on this as they claim that they are jointly using many common available data-sets.

**Léa Pousse-Beltran et al. (2020).** The 2020 M w 6.8 Elazığ (Turkey) earthquake reveals rupture behaviour of the East Anatolian Fault, AGU-*Geophysical Research Letters (GRL),* https://agupubs.onlinelibrary.wiley.com/doi/abs/10.1029/2020GL088136, also available at **ESSOAr** | https:/doi.org/10.1002/essoar.10502613.1, First posted online: Sunday **29 Mar 2020**.

Pousse-Beltan et al. (2020) also deal with the 2020 $M_W$ 6.8 earthquake, and its rupture properties by using satellite geodesy and seismology. They mainly investigate the mainshock rupture, postseismic deformation and aftershocks, and relations to previous earthquakes. According to their model, to the ENE the mainshock may have propagated into the rupture zone of the 1874 M ∼7.1 Golcuk Golu earthquake, and then stopped in the Lake Hazar basin, considered hosting a major EAF segment boundary. To the WSW the rupture propagated to the WSW at ∼2 km/s and halted after ∼20 s along a straight, structurally simple section of the Puturge fault segment. Furthermore, their study indicates bilaterally propagating rupture at relatively slow propagation speed from a nucleation point on an abrupt ∼10° fault bend. Their model suggests the mainshock rupture with a pronounced shallow slip deficit that is only partially recovered through shallow afterslip and they keep discussing further. However, there is no significant surface rupture observed at distinctive studies already reported.

Hence, outstanding and open questions are:

1. I have further noticed by closely analysing InSAR data that this more complex geometry is NOT necessary to fit the InSAR observations as Pousse-Beltran et al. (2020) accomplished two disconnected fault planes with different dip to fit the InSAR data. So, I wonder what is the opinion and/or explanation of the authors on this matter? Explain it in details as you both use similar type of data-set in order to help reader of wider geological community.

2. What are the major discrepancies among their findings and major results in the present work?

3. How and why do they interpret the overall results by using both seismology and InSAR data?

4. The authors should add through discussion on this article at Discussion/Conclusion sections.

**B.** Recently, Bletery et al. (2020) calculated a coupling map from InSAR and GNSS long-term velocities which suggests regions with slip deficit between 50-80% along the ruptured fault segment. Is there any further discussion and comments on this by the authors?

Bletery, Q., Cavalie, O., Nocquet, J-M., and Ragon, T. (2020). Distribution of interseismic coupling along the North and East Anatolian Faults inferred from InSAR and GPS data, submitted to AGU-*Geophysical Research Letters,* Earth and Space Science Open Archive (https://www.essoar.org/) Published Online: **Thu, 5 Mar 2020**, https://doi.org/10.1002/essoar.10502450.1.

**C.** I have also noticed the following article that refers to the 2020 Elazig earthquake, which is on line since **4 February 2020** under URL (https://eartharxiv.org/8xa7j).

**Jonathan R. Weiss et al. (2020**). High-resolution surface velocities and strain for Anatolia from Sentinel-1 InSAR and GNSS data. EarthArXiv Preprints, https://doi.org/10.31223/osf.io/8xa7j.

Weiss et al. (2020) claims that their "*3D velocity and strain rate fields illuminate deformation patterns dominated by westward motion of Anatolia relative to Eurasia, localized strain accumulation along the North and East Anatolian Faults, and rapid vertical signals associated with anthropogenic activities and to a lesser extent extension across the grabens of western Anatolia*".

I wonder how and why authors did not note/comment on this as they are also using assembled InSAR data-set in the Anatolia. Thus, I would like to hear what is the opinion and/or explanation of Jamalreyhani et al. on this matter? The authors should explain it in details as they both use InSAR data-set in order to help reader of wider geological community.

**D.** I wonder why the authors did not make use of the GNSS observations as they privilege (!) that they are using all the available data-set collected in the Anatolia.

**E.** Furthermore, I would like to see the *Finite-Fault Slip Distributions* on the preferred fault plane mechanism of the authors by using individual data-set, in pairs and with entire data-set that the authors have. For example, local data (the strong motion, AFAD?), and regional seismology data (KOERI, AFAD, GEOFON?) and teleseismic body-wave inversions may recover zones of large slip, while they are combined into a single large zone in the slip distribution by the geodetic inversion (InSAR? or GPS?).

The authors have only provided Finite-Fault slip-inversion jointly using InSAR and a few strong motion data. I am puzzled to see that they have not used available teleseismic and regional data-set? Then, we may continue debating in discussions and making resolved conclusions.

A new figure is needed on *Finite-Fault Slip Distribution* integrating below data-set separately.

(a) Teleseismic body waves (GDSN, FDSN, through IRIS DMC or else)
(b) Local seismic networks (AFAD, KOERI or else?)
(c) Regional seismic waveforms (AFAD, KOERI, GEOFON or else?)
(d) Strong Motion (AFAD, KOERI or else?)
(e) Geodetic (InSAR) (ESA, NASA or ALOS?)
(f) Coulomb (Cautious tidies work should be conducted)
(g) Seismicity (AFAD and/or KOERI?)
(h) Joint inversion with any of the above data-set to compare with each other.
(i) Full Inversion of all the above data-set.

I would like to see grid-space along-strike and along-dip with finite-fault slip distribution on these cells delineated with slip-vectors and displacement values (e.g. D-maximum, D-average), and evolution of seismic moment release as a function of time (i.e., source time function). This must not be too difficult to resolve and to retrieve over the inversion tools as there are much data.

**F.** The authors then can plot map-view of any of the above preferred ones on the morphology in order to compliment neotectonic and seismotectonics maps. Afterwards, we may then continue debating in discussions and making stable conclusions for likely future earthquakes in the region.

In conclusion, I still believe that this manuscript must have substantial major changes in the present form before further consideration for the *EGU-Solid Earth* as a brand-new submission.

Thus, it is NOT suitable and NOT acceptable for a publication at the *Solid Earth* journal as it is.

I recommend **REJECTION** and resubmission to SE and/or to any other journals.

---

## Referee Comment (RC3) · Anonymous Referee #3 · 17 Jul 2020

General Comments:

The coseismic data from some seismological networks and from SAR Sentinel-1 satellite are analyzed in order to estimate the fault parameters of the 24 January 2020 earthquake, understand the aftershock distribution, and the future distribution of events on the EAF. The paper is well structured and written. It represents an interesting application of mature software, with some interesting conclusions about the seismic gaps on the EAF fault. But, some conclusions and discussions are not examined with sufficient details, and some sentences are not completely debated. The time correlation among

the seismic events can be not studied (only) with an elastic model (Coulomb 3.3), but using also other types of models, for example, visco-elastic, visco-plastic. Some connection between the probable forecast events and the mainshock should be discussed with more detail, especially for the journal where the authors have submitted. The reviewer suggests acceptance after major revision.

Scientific Comments:

In the Introduction, the authors describe briefly the geodynamic context about the Anatolian plate and the East Anatolian Fault. The slab pull model and mantle flow model are only two of the several models discussed in the literature. For, example, the lateral extrusion of crustal wedges as discussed in Mantovani et al. 2001 (Short and long term deformation patterns in the Aegean-Anatolian systems: insights from space geodetic data (GPS) and Numerical simulation of the observed strain field in the central-eastern Mediterranean region) explain the kinematic of the Anatolian plate using a different point of view. I think, for the sake of completeness it is right to describe briefly and mention the other models of the Mediterranean geodynamic pattern. The paper represents an interesting application of mature software to analyze and inversion of seismic and SAR/GNSS data. Also, the authors use the Coulomb 3.3 software in order to estimate the coseismic static stress changes. The authors have developed and elastic model in order to estimate the spatial evolution of the Coulomb stress and they have discussed the correlation between the stress pattern and aftershocks distribution. Also, they have suggested that the increased stress in some parts of the EAF can expedite large earthquake activity in this region. I think that this elastic approach is a good model to understand the aftershock distribution, but to study the time distribution of the seismic events in an area it is necessary to use other models, for example, a visco-elastic model where the visco-elastic proprieties of the lower crust can be modeled and reproduce the time evolution of the stress field in the study area. I suggest to the authors introduce in the discussion and/or conclusion paragraph a brief discussion about the problems and limitations of the elastic model when are used in the earthquake

correlation time studies.

Technical corrections:

Line 44: . . ..it did not host major earthquakes during the last hundred years (Fig. 1): the most recent, large earthquake on the EAF dates to 1971 . . .. . . A more strong earthquake along EAF was 2010. . . ..

The 1971 event has occurred only about 50 years ago, and 2010 is only a few 'geological seconds' before now. It is not clear why the authors speak about the last hundred years. I agree with the authors that the large earthquake recurrence time on the EAF is greater than the NAF, but I suggest to the authors to modify the time span in these sentences in order to have an agreement.

Line 55: I think it is not completely correct to mention a paper only submitted.

Bletery, Q., Cavalie, O., Nocquet, J-M., and Ragon, T.: Distribution of interseismic coupling along the North and East Anatolian Faults inferred from InSAR and GPS data, Geophys. Res. Lett. Earth and Space Science Open Archive, https://www.essoar.org/doi/10.1002/essoar.10502450.1, submitted, 2020.

Line 69. Same consideration about a submitted paper. I think the mentioned results can be not reported.

Line 76: I suggest to the authors to use the same decimal digits about the Elazig-Sivrice earthquake (6.8) unless they have estimated the magnitude with associated uncertainty on the second decimal digit.

Line 116: Unfortunately, I am a physics, and if I write 6.77 $\pm$ 0.1 I do not pass the first exam of the Laboratory. I suggest to the authors to write 6.8 $\pm$ 0.1 and change in the text substituting 6.8 at 6.77. In Table 1, about this study are reported two 6.7 values, perhaps these values are 6.8.

Line 158: Most of the foreshocks including two Mw $\sim$5 are located very close to the

mainshock nucleation point, suggesting that they could have played a role in the main-shock preparation. This is a 'strong' sentence with support of only two references, but it can have important fallout, why the authors believe these earthquakes could have a role in the mainshock preparation, these events have anticipated or delayed the main-shock?

Line 187: Why do you use these values for Young, shear, and Poisson modulus? Line 188: I suggest to the authors to discuss briefly why they have chosen the middle value of the apparent coefficient of friction.

Line 192: In the caption of Figure S6 change Figure 3 with Figure 4 (I think)

Line 218: These loaded stresses can expedite future large earthquakes on either one of these segments. . . .. I think that the Coulomb stress has been estimated on the fault plane of the previous earthquake or . . ..  I suggest to the authors to explain in more detail these concepts.

Line 227: I suggest to the authors to indicate the Figure where the aftershocks cloud north of the EAF can be seen. I think it is the cloud at the NE near the lake.

Line 255: I suggest to the authors to report the three sectors discussed in Figure 3 in order to help the reader.

Line 264 and ...: I can in agreement with the authors about the increasing of the stress on some fault segments due to the study earthquake. The problem could be repre-sented that the elastic model adopted to give the 'instantaneous' stress increasing, as briefly discussed for the authors to provide the energy for the aftershocks. The possi-bility of a stress transfer could be investigated with viscoelastic or similar models where it is possible to model the distribution of the stress/strain in the time. But another ap-proach could require a lot of time, therefore I suggest to authors to discuss briefly the different approaches between elastic and viscoelastic (for example) models and the kind of results that they can obtain.

Line 264. . .. It is not clear in the text which scenario the authors believe it is more realistic (1, 2, or 4). Please clarify this point

Line 284: Local seismicity clusters, appearing months prior to the ElazÄśÄ§-Sivrice earthquake occurrence, probably track the slip instability onset. Probably I am in agreement with the authors, but they could briefly explain why these events have increased the stress on the Elazig-Sivrice fault. There is also a possibility that they have decreased the stress on the fault.

Line 525 Caption Figure 1: lost references about the kinematic pattern shown in the left up corner of the figure.

Line 675: lost reference about the active faults (Basili et al. 2013)?

---

## Author Comment (AC1) · 9 Sep 2020

**Reply Anonymous Referee #1**

Thank you for passing on the review of our manuscript on the Elazig-Sivrice earthquake and comments. We have taken great care to address all of the concerns. The detailed one-by-one response to the comments is included below (the review itself is in black, and our responses are in green).

The study presents a detailed analysis and interpretation of the 2020 January 24 Mw6.7 The Elazig-Sivrice earthquake. The study presents a full geodetic and seismo-logical analysis of the source rupture and the peripheral seismicity including foreshocks and aftershocks.
Coulomb stress and seismicity pattern – Interestingly, the south part of the fault has no aftershocks. This seismic quiescence is puzzling even more due to the prediction of the Coulomb stress analysis and the symmetry in the InSar data from both sides of the fault supporting that surface deformation took place at the south. Do you have explanation for that behavior?

The observation that the location of almost all aftershocks is north of the fault trace, is in good agreement with the focal mechanism solution and finite-fault modeling that point to an NNW dip of the fault plane. In a surface projection, an NNW fault dip offsets the rupture area at depth to the NNW. We show the centroid depth distributions of the aftershocks in Fig. 3, C1, C2, and C3.
The surface displacement measured by InSAR shows surface deformation on both sides of the fault, but at depth, the deformation will also be largest close to the rupture plane and again would show an offset to the NNW in a surface projection.
We have also updated the former figure 3 (Figure 5 in the new version) and figure 2 (Figure 3 in the new version) to show this result more clearly.
We also have added new sentences about the limitation of Coulomb stress modelling.

[Figure]

*Former Fig. 2, panel  c (Figure 3 in the new version): InSAR surface displacement maps covering the epicentral area. a) Masked and wrapped interferograms spanning the coseismic of the Elazığ-Sivrice earthquake: a) Ascending 21.-27. January, b) Descending 22. - 28. January). c and d) Ascending and descending subsampled surface displacements as observed, modeled and with the data residual. The grey filled box in c and d shows the surface projection of the modeled source, with the thick-lined edge marking the upper fault edge. The green star shows the epicenter of the Elazığ-Sivrice earthquake.*

[Figure]

*Former Fig. 3 (Fig. 5 In the new version). Spatiotemporal evolution of the 2020 Mw 6.8 Elazığ-Sivrice earthquake sequence (black stars always denote the mainshock, purple and cyan circles show aftershocks and filled brown circles show foreshocks). a) Spatial distribution of seismicity at the Pütürge segment, located between the Hazar Lake and the Yarpuzlu bend, showing the path of Firat River (blue line). Red lines show main faults (after Basili et al., 2013). Circles represent the epicentral locations of fore- and aftershocks (purple circles show 18 days of relocated aftershocks from Melgar et al., 2020 and cyan circles show AFAD catalog Ml 1+ and azimuthal gap less than 120°, last accessed 15 August 2020). The grey filled box shows the surface projection of the modeled source, with the thick-lined edge marking the upper fault edge. Focal mechanisms of the mainshock, 2 foreshocks and 19 aftershocks (focal spheres, color scale according to centroid depths) shown based on our moment tensor inversion. Black squares denote locations of the closest strong motion stations with their code. b) Depth cross-section along the profile DD' of relocated aftershocks (after Melgar et al., 2020) and events larger than Ml 4 (cyan) from AFAD catalog. The light pink rectangle shows the main rupture area and the dark vector shows the direction of the main rupture propagation, as resolved in this study. (c1-3) Depth cross-sections along profiles AA', BB' and CC', respectively (dip and width of all cross-sections are 90° and 20 km, respectively), showing the focal mechanisms of largest events (cross section projection). d) Temporal evolution of the aftershocks (Ml 1+ and azimuthal gap < 120°) versus longitude; the upper histogram shows the longitude versus the number of events N. The light yellow patch covers the longitudes ruptured in the Elazığ-Sivrice based on our finite-source modelling. e) Temporal evolution of the foreshocks (same style as panel d).*

Specific comments:
1- L. 24: Please use quantitative rather than "small" for the described foreshock cluster:

Thank you for this suggestion. We rephrased the abstract and manuscript, being quantitative. There are indeed two foreshock clusters, with maximum magnitudes of 4 April Mw 5.2 and 27 December Mw 4.9.
We have added new sentences in line 4 of the abstract and we removed "small" in line 24.

 "...Two foreshocks with Mw ≥ 4.9 and clusters of seismicity (Ml ≤ 3) located in the proximity of the main rupture's hypocenter..."

2- L. 29: Please explain how the statement for shallow locking depth corresponds with the seismicity range presented in Fig. 1 (0 – 30 km).

We believe that the instrumental and historical catalogs (Fig. 1) have a poor depth accuracy to discuss the shallow locking hypothesis by Cavalié and Jónsson (2014). Our statement is only based on the analysis of the Elazığ-Sivrice earthquake and supports this hypothesis. We have added the following sentences in the discussion section:

*"Estimated locations of historical and instrumentally recorded seismicity in the period 1900-2019, before the Elazığ-Sivrice earthquake sequence, show that they cover a large depth range from 0 to 30 km with no apparent pattern (Fig. 1). It seems changes of fault coupling with depth have a limited effect on the seismicity pattern. However the accuracy of the hypocenter depths is limited, particularly for the early times in this period, and maybe these locations lack the required depth resolution."*

3- L. 118: Please explain the usage of strong motion sensors to capture the low-frequency signal:

The low-frequency signals are used for the moment tensor inversion using regional and teleseismic broadband data. The resolved source geometry is then used to constrain the finite fault modeling. Here, we use higher frequencies (0.08-0.20 Hz) from strong motion sensors in the near field, to capture details of the rupture process. InSAR is important for finite-fault inversion to fix the lower-frequency image of the source and provides spatial resolution and constrains the fault position. Seismic data, especially near-field strong motion data is essential to resolve the temporal change in detail and provide better resolution (Anderson, 2003: Ide, 2007).

We have added the following sentences with references in the mainshock and method sections to clarify this issue and also the criteria that we selected the modeled stations:

In the mainshock section:

*"...Considering the signal-to-noise ratio, timing error, data availability, and azimuthal gap, we have selected six recordings of strong-motion stations of the AFAD network to be included in our optimization (Fig. 2). To capture the rupture process in space and time (Anderson, 2003; Ide, 2007), we use bandpass-filtered velocity records between 0.08 - 0.2 Hz "*

*and in the method section:*

*"Using combinations of different types of data-sets together helps to control different parts of the fault model (Ide, 2007). The InSAR data set together with the strong motions data allow to constrain the average slip, which can be less well constrained by either seismic or InSAR data alone (Ide, 2007)"*

*Anderson, J. G. (2003). Strong-motion seismology. International Geophysics Series, 81(B), 937–966.*

*Ide, S., 2007. Slip inversion. In: Kanamori, H. (Ed.), Earthquake Seismology. In: Treatise on Geophysics, vol.4. Elsevier, Amsterdam, the Netherlands. ISBN978-0-444-51932-0, pp.193–224.*

4- L.131: Did the earthquake rupture to the surface? This is not clear.

Thanks for pointing out the lack of clarity. The mainshock did not reach the surface with significant rupture. We see some disrupted fringes in the interferograms but these motions are very small compared to the average fault slip. Also, the modeled uppermost edge of the rupture at ~2.5 km depth, explains the lack of surface rupture. Now we have added new clear sentences as following in the mainshock section and with new references suggested by reviewer #2.

Furthermore, we have added the new sentence which mentioned the recently published study and their results about the absence of clear surface rupture (Pousse-Beltran et al. (2020)).

*"... no significant surface rupture is reported by the General Directorate of Mineral Research and Explorations of Turkey (MTA, 2020) nor apparent in optical satellite imagery (Pousse-Beltran et al., 2020). There is a pronounced slip deficit above the mainshock rupture, leading to, if any, very weak fault motion in some parts of the fault's surface trace (Pousse-Beltran et al., 2020)."*

*MTA. (2020). Preliminary field and evaluation report on 24 January 2020 Sivrice (Elazığ) Mw 6.8 Earthquake, General Directorate of Mineral Research and Explorations of Turkey (MTA), Ministry of Energy and Natural Resources, Ankara, 48 pages (https:// https://www.mta.gov.tr/).*

*L., Pousse-Beltran, Nissen, E., Bergman, E. A., Cambaz, M. D., Gaudreau, É., Karasözen, E., & Tan, F. (2020). The 2020 M*$_w$ *6.8 Elazığ (Turkey) earthquake reveals rupture behavior of the East Anatolian Fault. Geophysical Research Letters, 47, e2020GL088136.* *https://doi.org/10.1029/2020GL088136*

5- L. 206: It is not clear to me how did you conclude that the mainshock nucleated from the topper part of the fault plane from Fig. 3b. Please elaborate.

Thanks for pointing out the lack of clarity. In our finite fault modeling, we consider the fault as a rectangular plane. We optimized for the nucleation point, with variable locations on the faults along-strike and in depth. The finite fault inversion results show both that the rupture nucleated (1) at shallow depth and (2) ENE of the main ruptured area.
We apologize for the missing information on the resolution of the nucleation point and included a plot in Figure S13 to show the distribution of the hypocentral parameters. we provide the detailed output reports for all inversion runs in a separate online report at:
https://data.pyrocko.org/scratch/grond-reports/2020-elazig-sivrice/#/

We also clarify in former figure 3 (Fig.5 in the new version). They clearly show a shallow hypocenter of about 5 km, which is in the upper part of the fault plane with the confidence of 1 km (The uppermost edge of the fault plane is modeled at ~2.5 km depth and the width of the fault plane is ~9 km).

[Figure]

[Figure]

Missing information in the former figure S 13.

We have added the following sentences in the methodology and discussion sections to clarify these criteria. We also improve the former fig 3 (Figure 5 in the new version).

In the methodology section:
*"In the finite-fault modelling, we assume a planar rectangular rupture area with uniform slip, similar to the model by Haskell (1964). It is defined by 14 parameters: centroid time, three coordinates for position of the rupture plane, width, length, the two angles strike and dip, two coordinates for the nucleation point on the plane, slip rake angle, slip amount, rupture velocity, and rise time."*

In the discussion section:
*"Our results show a shallow nucleation point at about 5 km depth, which is in the upper part of the fault plane (Fig. 5), formally with a very small confidence interval of 1 km. Mai et al. (2005) showed that most strike-slip earthquakes nucleate near the bottom of the rupture plane. The Elazığ-Sivrice earthquake could be contrasting this tendency, but it is also possible that this result is an artifact of our inversion. From our experience with the nucleation point parameter from other examples, we think that a depth bias is likely due to oversimplification of the earth and source models."*

6- Fig. S5 – This is a very nice presentation of a unilateral rupture. I think it should be included in the main text. Please also consider presenting the seismic traces with azimuth to support the apparent duration measurement.

Thanks for the positive feedback. We agree to include this figure in the main text and we have added a new plot to also include the seismic traces.

[Figure]

*Figure 4: Mainshock apparent rupture durations (unit in seconds) at regional seismic stations. a) color-coded in map view. b) as a function of azimuth. c) Seismic traces used for apparent rupture duration sorted by azimuth.*

Best regards,
Mohammadreza Jamalreyhani
(on behalf of all co-authors)

---

## Author Comment (AC2) · 9 Sep 2020

**Reply Anonymous Referee #2**

We thank the reviewer for his/her many valuable comments. We identified a number of major questions, to which we reply. Following that, we report single answers to all the comments.
In the following document, the original review text is in black, our reply in green.

**1. Method and results**

From the reviewer letter, we recognized that the adopted methodology, resolving the mainshock and finite source parameters at different stages and combining the fit of different seismic and deformation data, was not clearly explained.

Reviewer comments indicate that the finite source inversion was not properly described, potentially leading to some misunderstanding. In our approach, we derive a simple rectangular finite source model with uniform slip. While we do not image slip heterogeneities, we believe that our approach is beneficial to robustly resolve first-order kinematic source parameters.

We use a combination of near-field strong motions along with InSAR surface displacement, to retrieve a kinematic finite source model with a uniform slip of the mainshock. The InSAR data set together with the strong motions data allow to constrain the average slip, which can be poorly constrained by either seismic or InSAR data alone. Using various types of data set together helps to control different parts of the fault model (Ide, 2007).

Although all used methods are described in previous publications (e.g. Kühn et al. 2020), we have completely reformulated the methodological section and we now provide an accurate description of the procedure to resolve point and finite source parameters.

*Daniela Kühn, Sebastian Heimann, Marius P. Isken, Elmer Ruigrok, Bernard Dost; Probabilistic Moment Tensor Inversion for Hydrocarbon-Induced Seismicity in the Groningen Gas Field, The Netherlands, Part 1: Testing. Bulletin of the Seismological Society of America doi: https://doi.org/10.1785/0120200099*

*Ide, S.: Slip inversion. In: Kanamori, H. (Ed.), Earthquake Seismology. In: Treatise on Geophysics, vol.4. Elsevier, Amsterdam, the Netherlands. ISBN978-0-444-51932-0, pp.193–224, 2007.*

**2. Source parameter uncertainties**

The reviewer repeatedly asks about source parameter uncertainty estimations. We apologize, as our approach on the data error propagation and model uncertainty estimation was not sufficiently clear in the earlier manuscript version. We use a bootstrap approach that follows a Bayesian strategy to estimate the model parameter uncertainties based on data error estimations providing data error variance-covariance matrices (Sudhaus and Jonsson, 2009) combined with Bayesian bootstrap weighting (Rubin, 1981). Data error estimates and Bayesian weights build a large set of different objective functions that provide

us with an ensemble of probable source models. From the distribution of these models we draw the uncertainties reported in the manuscript. We have improved the method section on uncertainty and the way they are estimated.

**3. Location of the seismic sequence**

The reviewer asks if there is potential misinterpretation of the seismic sequence, which locations are based on the AFAD seismic catalog. We stated that locations, as e.g. plotted in former Fig. 3, are absolute locations from this catalog and not relocations. We selected only the well-recorded events (azimuthal gap < 120 degree, a high number of records) among the catalog. However, our conclusions are mostly based on our own analyses and these catalog locations are only used as a reference to compare the overall distribution of seismicity with the finite source optimization performed in our study. In this way, we believe they are not overinterpreted even when their accuracy is lower than for a relocated catalog. Recent results of relocations, as published e.g. by Pousse-Beltran et al. (2020, GRL, Calibrated hypocentral relocations method, mloc package), Melgar et al. (2020, GJI, Double-difference location method, HypoDD package) and by the AFAD report (2020, HypoDD) show the major features discussed in our work, such as the seismicity distribution along the northern side of the EAF, with a larger offset in the SW section, compared to the NE section or less number of aftershocks in the main rupture area.
However, we have updated the early aftershocks catalog using relocated aftershocks by Melgar et al., 2020.

Diego Melgar, Athanassios Ganas, Tuncay Taymaz, Sotiris Valkaniotis, Brendan W Crowell, Vasilis Kapetanidis, Varvara Tsironi, Seda Yolsal-Çevikbilen, Taylan Öcalan, Rupture kinematics of January 24, 2020 *Mw 6.7* Doğanyol-Sivrice, Turkey earthquake on the East Anatolian Fault zone imaged by space geodesy, *Geophysical Journal International*, , ggaa345, https://doi.org/10.1093/gji/ggaa345

L., Pousse-Beltran, Nissen, E., Bergman, E. A., Cambaz, M. D., Gaudreau, É., Karasözen, E., & Tan, F. (2020). The 2020 $M_w$ 6.8 Elazığ (Turkey) earthquake reveals rupture behavior of the East Anatolian Fault. *Geophysical Research Letters*, 47, e2020GL088136. https://doi.org/10.1029/2020GL088136

**4. Data access to strong-motion data**

Local and regional strong-motion data has been accessed from the AFAD site shortly after the earthquake occurrence (24 Jan) and before Jan 27, when the website closed for maintenance. Data access was performed by email request, which was positively replied by AFAD. Apparently, other teams followed the same procedure in the same early days, as both broadband and strong-motion data were processed e.g. by Pousse-Beltran et al. (2020, GRL, broadband data) and Fountoulakis et al. (2020, EGU, strong-motion data). Data is now open again (https://tadas.afad.gov.tr/login and https://tdvms.afad.gov.tr/). The only study which claimed the absence of near/regional seismic stations for the Elazig-Sivrice earthquake is by Melgar et al. 2020. They likely attempted a download in the period of maintenance.

*Fountoulakis, I., Evangelidis, C., and Ktenidou, O.-J.: Imaging the rupture process of recent earthquakes using backprojection of local high frequency records, EGU General Assembly 2020, Online, 4–8 May 2020, EGU2020-13283, https://doi.org/10.5194/egusphere-egu2020-13283, 2020.*

**5. Figure quality**

The reviewer posed several comments regarding clarity, quality, necessity of our figures in the main and supplementary material, asking for improvement and/or to remove figures, which are considered unnecessary. We thank the reviewer for his/her thorough work. We agree with many suggestions and follow those, which improves the overall figure layout, readability, and information content. However, we have decided not to consider all suggestions on the figures, also taking into account comments and suggestions by the other two reviewers. For instance, figures S8-S13, which are all suggested to be removed, are important to illustrate the estimated model parameter uncertainties (uncertainties are indeed identified by the same reviewer to be an important topic for discussion). Therefore, we consider that most of them are important to complement our main results and we prefer to keep them as supplementary information. Thus, we provide the detailed output reports for all inversion runs. The optimization results of finite-fault modelling (InSAR and strong-motion), point sources of the mainshock (teleseismic and regional data) and fore- and aftershocks (regional data) together with model parameters uncertainties, parameter trade-offs, the Grond input configurations, and detailed output reports are available in a separate data publication at:
https://data.pyrocko.org/scratch/grond-reports/2020-elazig-sivrice/#/

We improved the layout of some figures according to comments. Comments to specific figures are provided below in the point-to-point replies.

Point-to-point replies

Abstract
The authors state that "2020 Elazig-Sivrice mainshock shows that the earthquake, with a magnitude Mw 6.77, ruptured at shallow depth (5±2 km) with a left-lateral strike-slip focal mechanism, with a dip angle of 74°±2° and a causative fault plane strike of 242°±1°, which is compatible with the orientation of the EAF at the centroid location". Why did they accept this solution of the 2020 Elazig earthquake? However, in text, they reported three diverse focal mechanism parameters from inversions with different data set (see the Table 1; strike/dip/rake/depth/Mo/Mw). But, in Abstract, they mixed all parameters. For example, they selected strike and dip angles of FF (InSAR and Strong Motion), but they selected focal depth and Mw from MT (with Regional and Teleseismic data). So, what is the scientific motivation for these choices in describing the focal mechanism parameters of the 2020 Elazig earthquake? Therefore, Abstract needs serious corrections in many ways.

We thank the reviewer for pointing out the missing clarity. We apologize for some typo. We first invert point source parameters using regional and teleseismic broadband data. These results are used to provide information for a more narrow and refined region of probable models to define the model space for the finite source optimization. Then all source parameters (e.g. rupture dimension, velocity, slip, nucleation point) were estimated, together with their uncertainties.

We improved the description of our procedure in the method section (see reply to main comment #1) and improved the abstract text: value reported here are the final results after the finite source inversion, except for the depth, which is defined as the centroid depth. We have edited the abstract and we report all results based on a finite fault model with presenting top edge depth of the plane.

*"The source model for the 2020 Elazığ-Sivrice mainshock shows that the earthquake, with a magnitude Mw 6.8, ruptured at shallow depth (top edge of the rupture at ~2.5 km) with a left-lateral strike-slip focal mechanism, with a rupture dip angle of 77°±2° and a strike of 241°±1°, which is reflecting the orientation of the EAF at the centroid location."*

In general, introduction, method and results sections (e.g., mainshock, seismic sequence, Coulomb failure stress change analysis) are NOT clearly presented and NOT well-written. For example;

A. The details of each methodology used in this work are not adequately explained here. The authors only described what they did in this study. No detailed information about the processing steps are provided on the selected algorithms (e.g., Grond, SNAP, Kite, Coulomb 3.3 etc.) There are too many unexplained sections about them making it difficult for the readers. They only directed the readers to the Pyrocko webpage for getting the information on each algorithm. However, I think it would be appropriate to present additional details clarifying each method and work devoted, since it is an important part of the article.

We have reformulated the methodological section. Please see main reply #1 and new submitted version of the manuscript.

Furthermore, the authors did not present their results in appropriate ways. The results section contains missing and incomprehensible parts. For example, on page 6, the authors reported that "We performed a moment tensor inversion for 18 earthquakes (2 foreshocks and 16 aftershocks) with Ml ≥ 4.3. For this purpose, we proceed as for the point source inversion. However, due to the weaker magnitude, we rely on only regional broadband data of the KOERI network (Fig. S3)". But they did not provide any BWIDC or surface-wave point-source MT results in Fig. S3 which only presents wrapped and unwrapped interferograms spanning the co-seismic of the Elazig earthquake for both ascending (up) and descending (down) directions. The authors should be very careful in numbering and referring to figures…

We apologize for some typos. We improved the text and figure citation accordingly.

B. Also, the authors provided time domain waveform fits for some selected traces for the best model at teleseismic (P-and SH-waves: tp.p and td.s) and regional (Rayleigh and Love waves: rd.rayleigh and td.love) distances. But they did not summarize the obtained/preferred source parameters (e.g., source mechanism models) from these waveform fits.
Hence, I think the results are generally not displayed properly and satisfactorily presented in Fig. S7.

The source parameters obtained from teleseismic and regional are provided in Table 1 and former figure 3.
We now have provided an online report for all optimization results.
Please see the main reply #5.

There are too many points to criticise and to question further such as:
1. Did the authors also cross-check P-wave first motion polarities recorded at near-field stations? Did they examine if the P-wave polarities are compatible with the nodal planes or not?
We believe a moment tensor optimization is more reliable than a first-motion polarity analysis, because the full waveforms are taken into account. The P wave first-motion polarity analysis can be affected e.g. by manual polarity pick, as emergent P waves arrivals make the picking challenging, and data filtering. We have estimated the robustness of our solution with the Bayesian bootstrap approach. We know we have reliable solutions based on these estimates. On the other hand, our result is in agreement with e.g. Pousse-Beltran et al. (2020) as well as global MT catalogs (Global CMT, GEOFON). We now cite Pousse-Beltran et al. (2020) in the manuscript.

2. The authors compared their focal mechanism solutions with those source parameters reported by other agencies (GCMT, GEOFON, AFAD etc., see Table S1). What are the main differences between these solutions? Did they check the effects of variations in each source parameters on wave form fits? Why do they suggest that their parameters are more reliable than the other solutions? Authors should verify and stipulate additional figures/plots, maps in order to convince readers that their results are significantly appropriate and better than the others.

All our estimates for source mechanisms are based on optimizations of source parameters such that we achieve a good fit of synthetic data to the observed ones. Our bootstrap approach enables the sampling of source models that produce data fits only insignificantly different from the best fit. So, yes, we tested the effect of different source models on the waveform fit. This information is given with the model parameter uncertainties. Our MT solutions (Table 1 and Table S1) are in good agreement with reference ones. For example,

the Kagan angle among our mainshock solution and the reference ones vary between 7.6° (compared to Global CMT) and 20.6° (compared to KOERI).

Furthermore the average Kagan angle for the aftershocks, when compared to reference solutions is ~30°.

All the information about our model is available in the manuscript and supplementary material, so that results can be reproduced.

See also main reply #5, and the following link for detailed output reports.

https://data.pyrocko.org/scratch/grond-reports/2020-elazig-sivrice/#/

3. How did the authors calculate the uncertainties of each source parameters? Which method was used to determine the amount of uncertainties in the source parameters? There is a crucial need to clarify these points (see page 4 lines 110-115).

We thank the reviewer for pointing out the missing clarity in the method part. Please see also our general reply #2 above. The uncertainties are estimated based on a Bayesian bootstrap approach. The misfit contribution of each waveform is weighted differently in 100 parallel optimizations. In this way station noise, and other observational errors will affect the final solution of each parallel optimization. The scatter in the final model parameters reflects the influence of these observational errors and allows the model uncertainty estimation. Please also see the main reply #1 and #2.

4. Why did the authors select the frequency band of 0.08-0.20 Hz in the finite source optimization/inversion with near field data? Similarly, the authors modelled entire waveforms and amplitude spectra in the frequency band ranging 0.02-0.05 Hz. How and why did they select these frequencies?

In our procedure, we first resolve the point source parameters fitting low-frequency data at regional to teleseismic distances using a point source approximation. Next, we optimize a finite source, assuming a planar, rectangular source, which can be constrained using near field data and higher frequencies (0.08-0.20 Hz) , which is a usual procedure (see also main reply #1).

For the aftershock focal mechanism we used frequency bands ranging 0.02-0.05 Hz. These low-frequency seismograms, below the corner frequency, contain enough information to retrieve point source parameters for this range of magnitudes (e.g. Cesca et al. (2010)).

We have added a new sentence with references in the mainshock and aftershock sequence sections to clarify this issue.

In the mainshock section:

*"To capture the rupture process in space and time (Anderson, 2003; Ide, 2007), we use bandpass-filtered velocity records between 0.08 - 0.2 Hz."*

In the seismic sequence section:

*"These low-frequency seismograms, below the corner frequency, contain enough information to retrieve point source parameters for this range of magnitudes (Cesca et al. 2010)."*

*Cesca, S., Heimann, S., Stammler, K., Dahm, T. (2010): Automated procedure for point and kinematic source inversion at regional distances. - Journal of Geophysical Research, 115, B06304. https://doi.org/10.1029/2009JB006450*

5. Is there any slip distribution/rupture propagation model with the amount of displacements on the fault plane for which the authors favour? How did they estimate the fault length and fault width for this earthquake? They barely provided some waveform fits that are not clearly recognised (see comment E on Discussions and Conclusions).

We assume a rectangular source model with a uniform slip. Heterogeneous slip models have a much higher degree of freedom, e.g. a free slip and rake per subfault. We keep our fault model more simple and estimate in a fully non-linear way the first-order characteristics of the fault and the dynamic source parameters. The text in the method section has been improved to better explain our procedure (see also reply to major comment #1).

6. They also did not evidently explain that which source parameters (Table 1or else?) were used in Coulomb stress change analysis. They only mentioned that a homogeneous elastic half-space Earth model and the causative fault of the Elazig-Sivrice earthquake are considered as a rectangular dislocation (26 km long 9 km wide with mean slip equal to 1.8 m) in Coulomb stress modelling. What is the causative fault plane of this earthquake? The authors should distinctly summarize each earthquake source parameters resolved (e.g., strike/dip/rake angles/seismic moment/depth etc) that they used in Coulomb stress analysis. Furthermore, the authors should explain how they calculated the mean slip of 1.8 meters given in this section too (see Page 6 line 190).

We clarify this part and now explicitly included the information on the geometry of the causative fault plane for the Coulomb stress modeling, which is the one derived by the finite source inversion.
The finite source optimization resolves the ENE-WSW orientation of the fault plane, which is also in obvious agreement with the EAF orientation. We use uniform slip in our source model, and this estimated slip is used in the Coulomb stress change calculations. We have added this sentence in the Coulomb stress section:

*"… the causative fault of the Elazığ-Sivrice earthquake is considered as a rectangular dislocation based on our obtained finite-fault model presented in Table 1 (27 km long, 9 km width, mean slip equal to 1.8 m, 241° strike, 77° dip, and 0° rake)."*

7. How did the authors estimate the rupture duration and rupture velocity? (see page 5 line 135).

We have estimated the rupture velocity, dimension, and directivity by the finite source optimization, together with their uncertainties. The rupture duration of ~27±1 second is calculated based on moment tensor inversion.

We extended the method section to make this explicit.

See also next comment.

C. Nonetheless, my biggest concern is primarily on how the authors describe the rupture propagation and its time evolution in the current manuscript. Neither in the text nor in the supplementary material they present convincing material documenting the signature of unilateral propagation that the authors generously claimed. I would expect to see spatio-temporal co-seismic slip behaviour following the inverse modelling of such data set, if any. However, this is a very critical detail regarding the physics of time evolution of this earthquake, and there is NO evidence in this current work to clarify and/or to debate on these diversities of observations and interpretations.

In our finite source model, the rupture propagation is represented with a constant rise time (fixed value of 1 sec) and a constant rupture velocity in the whole rupture zone (free parameter in the optimization). The rupture starts radially from a nucleation point, the position of which is also free in the optimization. Whichever configuration produces a good data fit is tested during the optimization and we found that a nucleation in the north-east, close to the edge of the fault, best reproduces the observation. This agrees well with the independently confirmed hypocentral-centroid relative location, the azimuthal pattern of apparent rupture durations, the spatial distribution of saturated broadband stations at local distances and also azimuthal pattern for the PGA of near field strong-motion stations. We update the former figure 3 to show the rupture propagation more clearly.

[Figure]

*Former Fig. 3 (Fig. 5 In the new version). Spatiotemporal evolution of the 2020 Mw 6.8 Elazığ-Sivrice earthquake sequence (black stars always denote the mainshock, purple and cyan circles show aftershocks and filled brown circles show foreshocks). a) Spatial distribution of seismicity at the Pütürge segment, located between the Hazar Lake and the Yarpuzlu bend, showing the path of Firat River (blue line). Red lines show main faults (after Basili et al., 2013). Circles represent the epicentral locations of fore- and aftershocks (purple circles show 18 days of relocated aftershocks from Melgar et al., 2020 and cyan circles show AFAD catalog Ml 1+ and azimuthal gap less than 120°, last accessed 15 August 2020). The grey filled box shows the surface projection of the modeled source, with the thick-lined edge marking the upper fault edge. Focal mechanisms of the mainshock, 2 foreshocks and 19 aftershocks (focal spheres, color scale according to centroid depths) shown based on our moment tensor inversion. Black squares denote locations of the closest strong motion stations with their code. b) Depth cross-section along the profile DD' of relocated aftershocks (after Melgar et al., 2020) and events larger than Ml 4 (cyan) from AFAD catalog. The light pink rectangle shows the main rupture area and the dark vector shows the direction of the main rupture propagation, as resolved in this study. (c1-3) Depth cross-sections along profiles AA', BB' and CC', respectively (dip and width of all cross-sections are 90° and 20 km, respectively), showing the focal mechanisms of largest events (cross section projection). d) Temporal evolution of the aftershocks (Ml 1+ and azimuthal gap < 120°) versus longitude; the upper histogram shows the longitude versus the number of events N. The light yellow patch covers the longitudes ruptured in the Elazığ-Sivrice based on our finite-source modelling. e) Temporal evolution of the foreshocks (same style as panel d).*

In fact, if there is a proposed model of co-seismic slip distribution based on the inversion of InSAR data set, I have not seen any relevant model result and I am wondering why authors avoided to share these details, if any. The absence of segmentation is only mentioned very briefly referring to the InSAR data modelling in Fig. 2c (Page#8, Lines#: 245-247, see: "Some systematic residuals in the near-fault InSAR results of the finite slip modelling (Fig. 2) may point to a slight segmentation, but the overall good data fit in the single-segment finite fault modelling suggests that segmentation is not a first-order feature."). But unfortunately, I cannot see any clear elaboration from the interpretation of Fig.2c. Even it is exceedingly unclear to what these two different InSAR modelling results belong which specific data subset. At this stage answer to this issue is very critical because the reader can have tough times in understanding the link between the rupture process and InSAR data with only available information presented in the current form of the manuscript. It would be nice to see snap-shots and a movie of time evolution on the preferred model of co-seismic slip distribution using InSAR data-set.

We apologize again for lack of clarity on the finite rupture model, which is here a uniform-slip model. The parameters of this model are provided in full and the homogenous slip model now sketched in Fig. 3.
The near-field data, in particular InSAR data, do not indicate a strong rupture segmentation and the modelled synthetic data fit well enough to the observation without doubling the degree of freedom by allowing for two segments instead of one. With more free parameters the fit to the modelled data is quite naturally increasing, but if the increase in data fit is significant enough would need an extra analysis, e.g. based on informational criteria by Steinberg et al., (2020). We extend this explanation in the discussion to be more clear.
*Andreas Steinberg, Henriette Sudhaus, Sebastian Heimann, Frank Krüger, Sensitivity of InSAR and teleseismic observations to earthquake rupture segmentation, Geophysical Journal International, , ggaa351, https://doi.org/10.1093/gji/ggaa351*

As for the modeling of InSAR data (former Fig 2c, now Fig.3), these are not two models, but the same model using ascending and descending orbits. The figure and its caption has been improved to make this more clear.

[Figure]

*Former Fig. 2, panel c (Figure 3 in the new version): InSAR surface displacement maps covering the epicentral area. a) Masked and wrapped interferograms spanning the coseismic of the Elazığ-Sivrice earthquake: a) Ascending 21.-27. January, b) Descending 22. - 28. January). c and d) Ascending and descending subsampled surface displacements as observed, modeled and with the data residual. The grey filled box in c and d shows the surface projection of the modeled source, with the thick-lined edge marking the upper fault edge. The green star shows the epicenter of the Elazığ-Sivrice earthquake.*

D. Another surprising issue for me is that why authors did not consider a direct strong motion data analysis obtained from a fairly dense station distribution though this is also questionable issue (?) as the Turkish Government AFAD authority officially released the data set onthe16 June 2020, and not before! Hence, I am NOT contented how the authors attained this data-set which was not available on the days of 24-25 January 2020 and afterwards until the 16thJune 2020.

Thus, I am quite curious how the authors obtained these unreleased strong-motion data which should be clarified and confirmed in writing from the Turkish government authorities. Otherwise, this does NOT grant an equal opportunity on Data Availability for international scientists to conduct a research on the current and other relevant earthquakes in the region for global and/or regional mutual interests. Therefore, I consider this current work being NOT an objective piece of scientific conduct, and it is quite unfair to the others interested to study these events further.

Furthermore, under these conditions, one would investigate the time variation of the pseudo displacement that could be easily extracted from these near-field recordings in relation to the ground motion using these stations located at different azimuths with respect to the mainshock. This, therefore, would give a direct and less-biased information on likely different episodes of the propagation. The authors should revisit these issues to elucidate further. And, it would be healthier to see snap-shots and a movie of time evolution on the preferred model of co-seismic slip distribution based solely on strong-motion data-set.

We strongly refute this criticism.
Strong-motion data have been used for finite source optimization (Uniform slip model) as we explain in the mainshock section.
The Strong-motion data were open early after the mainshock until the AFAD closed its website on 27 Jan 2020 for maintenance. Data were downloaded at that time and now are again available (see reply to main comment #4).

E. Page# 8, Lines#: 243-245, "*Scenario 3 is unlikely, because the depths estimated by earthquake relocation and aftershock centroid MT inversion do not change significantly along the fault. Both scenarios, 1 and 2 are plausible and in contradiction with the observed data*"
I do not think that the AFAD's routine PDE locations have very high resolution that can enable us a precise aftershock distribution to comment on further. It is pretty clear that they are not quite reliable to make such firm conclusions. The biggest problem with their relocated earthquakes stems from the very irrelevant type of 1-D initial seismic velocity structure model used in their localization procedure. Thus, the best option would be to perform relocations based on conventional relative techniques since they can provide much precise values as they do not depend on presumably uncertain knowledge of seismic velocity structure. Relative locations, for instance, HypoDD will better work in keeping track of the spatio-temporal behaviour of the seismicity much consistent. Precise relocation is highly achievable via phase reading data set that is publicly accessible dataset from the AFAD and other regional archives of KOERI or so (see: Waldhauser F. and W.L. Ellsworth, 2000. A double-difference earthquake location algorithm: Method and application to the northern Hayward fault, Bull. Seism. Soc. Am., 90, 1353-1368, 2000; Waldhauser, F., 2001. HypoDD: A computer program to compute double-difference earthquake locations, USGS Open File Rep., 01-113, 2001).

Actually the sentence is: "*Both scenarios, 1 and 2 are plausible and NOT in contradiction with the observed data*". See the line 244 in the original manuscript.

Relocation is beyond the scope of our study. However, the AFAD catalog is now providing also relative locations and Pousse-Beltrans et al. 2020 and Melgar et al. (2020) also relocated the aftershocks. These relocations do not change the picture and do not contradict our arguments. However we have updated the aftershocks using relocated aftershocks by melgar et al., 2020.
See also the main reply #3.

On the other hand, foreshocks can be often described as small event activities in a close proximity to the hypocenter of the mainshock. They may have an essential role in understanding the physics and initiation process of an upcoming event. Two main models have been, so far, proposed to explain the link between foreshocks and the main rupture. These are pre-slip (Ellsworth and Beroza, 1995) and cascade models (e.g. Fukao and Furumoto, 1985). In order to efficiently evaluate whether a series of events can be regarded as the foreshock activity and the proper mechanism involving the type of physical process affecting on the fault plane requires a tedious and critical investigation on extremely precise event localization, spectral analyses for their high resolution source characteristics (e.g. source radius, released energy, etc.), or the amount of stress change they caused. Although, the present work refers to possible foreshock activities at few places in the text, it is hard to see reasonable arguments if these activities can be interpreted within the concept of foreshock classification. There are almost no detailed efforts performed in the present work to elucidate this issue.

Thank you for pointing out this. We refer to the "foreshock" activity, due to the similarity of location, focal mechanism, and waveforms, as reported at L. 274-276 (original manuscript). These events have been identified as foreshocks by Pousse-Beltran et al. (2020). We consider relevant to report on such localized seismic activity preceding the mainshock because this type of observation has been done previous to other large earthquakes along with the EAF (2010 Mw 6.1 Elazig Earthquake (Tan et al. 2011)), which is now cited with respect to the observation of foreshock activity.

Moreover, we removed the part of the discussion about the role of these foreshocks in mainshock preparation according to the suggestion of reviewer 3 (See reply to the reviewer 3).

**Figures:**

Most of the figures are NOT well prepared for a clear publication quality. They are rather busy, and there must be a way to make them look easier to read. The fonts used on the maps and seismograms (see Fig. 2a, b, c; Fig. S7, S11, S13, S14, S15) are too small, hence it is hard to read. There are also many missing sections and references as such in the figure captions. The authors should cautiously arrange/plot the figures and re-write the figure captions without leaving any open questions as far as copyright issues are concerned.

We rearrange the figures and we have added the missed references in the captions. See the point-to-point reply in the following:

Fig. 1 and Inset: The Inset is very confusing as it does not present right geometry of the NAFZ in the Sea of Marmara, nor the extensional features in the western Turkey and the Aegean Sea. There is immense amount of high-quality papers, graduate thesis and extensive geophysical and geological experiments conducted in the Sea of Marmara and the Aegean regions, but I do NOT see them noted in discussions or cited in references leaving many open

questions. What are the sources of historical and instrumental earthquake data and GPS velocity vectors plotted in Fig. 1? Any references to add?

We improved the figure 1 caption with references.

The inset has the main aim to illustrate the overall plate boundaries and main faults. NAFZ, Sea of Marmara and Aegean region are marginal to the paper topic. Main features have been plotted upon Bird (2003) and we consider this sufficient for the inset. The main part of Figure 1 reproduces more accurate fault segments after Duman and Emre (2013). We complete the list of used references in the caption, including Bird (2003) for the inset and specific ones for the historical and instrumental seismicity in the main figure (these were previously cited only in the manuscript, but not in the caption).

*Bird, P. (2003), An updated digital model of plate boundaries, Geochem. Geophys. Geosyst., 4, 1027, doi:*10.1029/2001GC000252*, 3.*

[Figure]

**Figure 1.** Inset: Simplified regional seismotectonic of the Anatolian block and escape tectonics. Black lines show Plate boundaries (after Bird, 2003). The plotted GPS velocity vectors are based on McClusky et al., 2000 et al., 2006. a) The EAF and segmentation are based on Duman and Emre (2013). Seven segments on the main strand of the EAF from west to east; the Amanos, Pazarcik, Erkenek, Pütürge, Palu, Ilica and Karliova segments. All historical and instrumental seismicity during 1900-2019 (before Elazig-Sivrice earthquake) and 4<=Mw<6.5 are shown as blue circles, and their frequency within the seven different segments illustrated by red bars in the inset. Among these, the large historical earthquakes (Mw 6.5+) are given with labels. Historical earthquakes are based on Ambraseys, 1989, Ambraseys and Jackson, 1998, Nalbant et al., 2002 and Duman and Emre, 2013. Instrumental earthquakes are based on AFAD and Kandilli catalogs. Black star shows the epicenter of the

Elazığ-Sivrice earthquake. Red lines indicate active faults in the region (Basili et al., 2013). b) The cross-section shows the depth of events with 4<Mw<6.5 during 1900-2019 along the EAF.

Figs. 2a, b,c;S2 and S4: In Fig. 2a, the axis information (i.e., coordinates of the location map) is too small and not readable at all. The readers cannot easily recognise which stations are in the rupture direction and which are in the opposite direction to the rupture zone (see Fig. 2a) as the names/codes of stations are not properly presented on a focal sphere (i.e.: azimuthal coverage), and/or on the map view (see Fig. S2). In addition, the number of seismic stations identified on the map (see Fig. 2a) is much more than the number of waveforms modelled. Explain the main reasons why the other stations are not modelled in the joint inversion. How did you select the modelled stations? What are the criteria? The authors should provide a summary Table specifically to elucidate Fig. 2a, and it can be posted to the supplementary.

We improved the layout of Fig. 2a to make all labels readable.
We believe the relative station location and azimuth can be appreciated on the map.
PGAs are reported for all stations, but good quality data (i.e. avoiding those with low signal-to-noise, timing error, data gaps, no data availability…) is available for 6 stations with 3 components. Furthemore the Radial components of stations with codes 4404 4407 and also the Transverse component of stations with codes 2105 and 4407 was not used due to the poor quality of the record.
We now have changed the figures and we show the strong motions waveform with stations distribution together in separate figure (new Fig. 2).
We also have added new sentences about the criteria that we selected the modelled stations:
"… *Considering high the SNR ratio, timing error, data availability, and azimuthal gap, we have selected 6 strong motion stations in our optimization (Fig. 2)."*

An additional table could only additionally provide reported PGA values, but these are already available in AFAD (2020), which we now cite in the figure caption.

[Figure]

Figure 2. Near-field data coverage (strong motion stations) in the epicentral area for the modelling of the causative fault plane of the 24 January 2020 Mw 6.8 Elazığ-Sivrice earthquake.  a) The squares show strong motion stations colored according to peak PGA values based on the AFAD report (AFAD, 2020). Stations with codes are those used in the finite-source optimization combined with InSAR data (Fig. 3). The dashed black boxes indicate the spatial extent of the used Sentinel-1 imagery from both ascending and descending orbits (Fig 3a). The red star shows the epicenter of the mainshock. Red lines indicate active faults in the region (Basili et al., 2013). The focal mechanism results shown are based on the joint inversion of teleseismic and regional datasets obtained in this study (double-couple component shown). The purple box shows the spatial coverage of Fig. 3b. b) Strong motions modelling: The best-fitting model in the Z component of six near-field strong motion stations;

observed trace (dark gray) and synthetic trace (red). Information in the waveforms fit (left side, from top to bottom) gives station name with the component, distance to the source, station azimuth, starting time of the waveform (relative to the origin time).

Fig.2b is not helping readers to identify the waveform fits, and details of the methods and software should be briefly summarized in the Supplementary On-Line material to guide those who are not familiar with this code. Besides, the choice of coloured lines and envelope is not helpful, therefore there is a need for further upgrade here.

Fig. 2b shows a selected example of waveform fits (vertical components) and is complemented by former Fig. S12. We improved its caption to describe all features. We also improved the methodological section, as suggested (see reply to main comment #1).
We also provide the detailed output reports for all inversion runs in a separate online report at:
https://data.pyrocko.org/scratch/grond-reports/2020-elazig-sivrice/#/

Fig.2c requires some more details regarding the difference between two different solution. What is the major observation, interpretation and discrepancies between top and bottom InSAR maps in Fig. 2c?
We improved the caption to clarify that Fig. 2c reports two different images (ascending, descending) and not two different solutions. All data in Fig. 2a, 2b and 2c are modeled with the same finite fault model.

Figs. 3. and S1: The entire Fig. 3 is NOT acceptable at all. It should be seriously revised. The aftershock distribution in this figure does not make any sense due to too much uncertainty inherently existing in the epicentre/hypocentre location data taken from the AFAD PDE data catalogues. I strongly suggest that the imperative data must be revisited by using relative relocations and/or any other conventional methods.

We strongly disagree on this point. Relocation is beyond the scope of our study. However, Pousse-Beltrans et al, 2020 and Melgar et al, 2020 also relocated the aftershocks. These relocations do not change the picture and do not contradict our arguments. We have updated the former fig 3 using relocated early aftershocks by Melgar et al., 2020.
Please see the main reply 3.

The authors should add a proper reference of the velocity model to the caption of Fig. S1, and also give right reference to the AFAD for earthquake locations plotted in Fig.3.
We apologize for the missing information and have included all the needed references to captions: AFAD for the locations in Fig. 3 and Acarel et al. (2019) and Bassin et al. (2000) for the velocity model in the updated Fig. S1.

The discussion based on these maps and figures are irrelevant, and they should be removed in text as it does NOT reflect the ground truth until authors improve them with the accurate relocation techniques.

Since recently the locations of seismicity have been confirmed by newly published earthquake relocation results of the sequence (e.g. Pousse-Beltran et al. 2020). We think this confirmation satisfies the point made by the reviewer and is no more of concern. We have extended the seismic sequence section to include these new findings with the references. We have updated the former fig 3 using relocated aftershocks by Melgar et al., 2020. Please see the main reply 3.

How the rupture area (red rectangle) and rupture direction in Fig. 3b are defined, likewise light-yellow coloured region in Fig. 3d,e? Is it simply based on inadequate seismicity map of the AFAD and/or KOERI or else, if any?

We apologize for the lack of clarity in the figure and caption. The rupture area represents the slip area of our finite fault model. The figure and the caption has been updated accordingly. The light-yellow area shows the mains rupture area. For the inadequately of seismicity map, please see main reply #3.
We have update the former fig. 3, we make the rupture area and direction more clear and we include the Rev #3 suggestions as well.

Fig. 4: As long as the aftershock distribution data is updated, presenting the vertical extent of the stress change along the fault plane can be physically more meaningful as well. The authors should add a proper reference for active faults plotted in the epicentral area. This figure should be presented in the supplementary along with other Coulomb stress change map in Fig. S6 with additional brief information on CST and up-to-date worthy referencing.
We thank the reviewer for this suggestion. For aftershock distribution please see the main reply #3. We have added two paragraphs in the Coulomb stress section accordingly also based on suggestions of reviewer #3 (see reply to rev #3 about the limitation of Coulomb stress failure). We have added a reference to the European fault database in the figure caption (black lines show the main faults, after Basili et al., 2013). We also have removed fig. S6 from the supplementary material.

Table 1: Source mechanism solutions are summarized, but I am still not quite sure how the authors claim that the magnitude of mainshock is Mw 6.77. Where is it taken from? How about the errors? How did you calculate them? The authors should provide parameters of their own results in a separate Table. There is NO introductory information regarding these material as the authors should add a cover page describing individual plots. It looks quite clumsy as a grab bag in its present form.

Table 1 reports the results of our own inversion (both point source moment tensor and finite source models) in comparison to other reference solutions. Our results are reported, for each listed parameter, in terms of mean value and uncertainty (see main reply #2). While we

have estimated the magnitude in our study with uncertainty, it is about 0.1. The value of Mw 6.77 is obtained for the best solution using all available data. We improved the text and table to state this clearly. We have changed all the magnitude to 6.8±0.1, also in the title. We fixed the wrong numbers in the table.

Fig. S1: What is the source for the 1-D radial velocity models used for calculating near-field Green's functions? Any references? Is Acarel et al. (1996) right one to refer to? According to my recollection, this region is characterized by relatively thick crust compared to the rest of Turkey and Moho depth taken about 31-32 km in this model could be misleading. I highly recommend authors to check recent papers dealing with Anatolian crust based on ambient noise tomography (e.g., Delph et al., 2015), Pn tomography tomography (e.g., Mutlu and Karabulut, et al., 2011), receiver functions (e.g., Vanacore et al., 2013; Karabulut et al., 2019) in order to obtain a reliable 1-D velocity model, among many other tomography studies in 1-D and even furthermore in 3-D FWI studies.

We use Acarel et al. 2019 (not Acarel et al. 1996!), which has been proposed for the study region, to model local data (i.e. strong motion and local surface deformation). We also use a regional crustal model (Bassin et al. 2000) and a global AK135 model (Kenneth et al. 1995) for fitting regional and teleseismic broadband data respectively. We have added a new figure beside the fig. S1 to show the regional crustal model (Bassin et al. 2000).
The Acarel et al. (2019) model has been used also by other authors for the same earthquake (Pousse-Beltran et al. 2020).
The suggested reference by Vanacore, Taymaz, and Saygin (2013) does not actually show clear evidence for a thick crust at the earthquake location. Moreover this would also have little influence in the modeling of near field effects for a shallow earthquake.

[Figure]

Figure S1. The velocity model for calculating Green's functions. Left: Acarel et al., (2019) for near-field, right: CRUST2.0 model for the regional distances (Bassin et al., 2000).

Fig. S2: Here, the distribution of teleseismic stations are plotted without station codes. Thus, it is making even more difficult for readers to identify waveform fits. Besides, there is a huge azimuthal gap in the North and North-East quadrants spanning from Greenland to Kamchatka peninsula? This is quite important especially when the authors speak about directivity of the main fault based on their observation at other data-set they claim that they have along NE-SW striking geometry. How can we see these propagation effects (i.e., doppler-shifting) in waveforms if we do not have stations at these azimuths? And, also, in SW azimuths. The authors should provide complete list of stations for which broadband P-waveforms obtained along with complete catalogue information in a Table. Additionally, arcs of latitudes and longitudes should be plotted at each 15 arc-distances or so in order to help readers for an orientation of the nodal planes.

Exact station codes, the corresponding azimuths and distances to the epicenter are fully reported in the subplots of Fig. S7, which shows the waveform fits - in text though, not graphically. The main function of Fig. S2 is to illustrate the station distribution to assess the station coverage in distance and azimuth. Showing 76 stations, it cannot be easily improved to show all station labels, but would rather become too busy in such an attempt.
The figure reports location of broadband stations at teleseismic distances, which were used in addition to regional stations to perform a moment tensor inversion. We have plotted new figure with arcs of latitudes and longitudes at each 15 arc-distances, as suggested. Now it is easy to follow the fit of a specific station with azimuths and distance which reported in the waveform fits.

[Figure]

Figure S2. Distribution of 76 teleseismic broadband stations at distances of 30-80 degrees (red triangles) used for moment tensor inversion. Focal mechanism of the 24 January 2020 Mw 6.8 Elazığ-Sivrice earthquake. The mechanism coordinate denotes the epicenter.

The amount of data used, with 76 stations at teleseismic distances and 12 stations at regional distances is large and allows a robust inversion, even in presence of a marginal gap to the NE in the teleseismic dataset. Suboptimal station distributions are indeed causing larger uncertainties in the moment tensor estimation and for a fact we have an azimuthal gap in the north of ~40 degrees. As we outlined in our reply above and as we now better explain in the manuscript, we conduct a thorough model uncertainty estimation to show the precision of our results. They are given in Table 1 and show that the mechanism, for example, has a 68% confidence for the strike, dip and rake of +/- 9 degrees each. These uncertainties reflect the precision possible under the given station distribution.

We may not be clear enough in the manuscript, but the rupture directivity is not studied using the teleseismic data. We analysed the rupture directivity and finite fault model using Strong-motion and InSAR. On the other hand based on regional seismic data (An independent analysis, indirect observation) we obtain the apparent duration that show stations in the front of the rupture direction show less apparent duration than those are located in backward of the rupture direction (See line 136 of first version of manuscript) and support our finite-fault results based on join inversion of strong motion and InSAR. Moreover, special distribution of the saturated stations, and Azimuthal pattern of the PGA (Former fig. 2) also support the rupture directivity (See also reply for comment on Fig. S4). Please see also main reply #3.

Fig. S3: The more detailed information should be provided in figure caption for wrapped and unwrapped interferograms of the InSAR data presented in Fig. S3.
We apologize, as this information was only provided in the main text. We have moved this figure in the main text and rewrite the caption with more information.

Fig. S4: This is a confusing map...Firstly, traces and/or outline of main fault zones in Turkey, in the Aegean and nearby countries are NOT precise and misleading as presented in inset of Fig.1. The authors should be very careful in cross-referencing others' data without proper knowledge. This figure ought to be replaced with right one as there are many experts around to give help. Otherwise, copy and paste fashion can be very damaging one, and it looks as if data-base of the main faults of Basili et. al. (2013) is NOT the right resources to make use of it...? Hence, remove this reference and use a decent relevant one on the active faults of Turkey and surroundings. Secondly, what are the sources and which networks of coloured triangles refer to? Green? and Red? triangles stand for which network? Are there misinformation here?
How about AFAD broadband stations as the authors were able to get some of their other type of data-set? Why those local and regional broadband seismic stations of AFAD are not used in joint inversions? Any clarification and/or explanation?

We apologize for not referencing the source of the plate boundaries that are actually after Bird (2003) in the caption. We do now. Triangles are used to denote broadband seismic stations, independent of the network, and the color only reflects whether data were saturated (red) or not (green). We improved the caption to make this clear.

The AFAD broadband stations were not used for the point source inversion, because (1) data were only available for the mainshock and few early aftershocks and (2) data quality and sensor orientation analysis was not available for this data. We make this now clear in the seismic sequence section:

*"The data quality of these seismic networks and errors in metadata have been evaluated. This analysis is so essential in waveform-based studies, especially in MT inversion (Peterson et al., 2019). The AFAD regional broadband stations were not used for the point source inversion, because the data quality analysis is not available for this network."*

Fig. S5: It would be better to mark the units of the colour scale-bar (s stand for seconds?) given next to the plots?? Above all I find this figure not a helpful one.

Yes, thank you for remarking on this! "s" stands for seconds as we clarify now in the caption. We follow EGU-Solid Earth policies, which encourage the usage of SI units.
The overall evaluation of the figure is in contrast with the comments by Reviewer #1, who suggests having it in the main document. We believe this result is important to discuss rupture directivity and extend the visualization of results following rev. #1 recommendations (see reply to rev #1, last comment).
We have also updated this figure to make it clear.

Fig. S6: This figure should be presented along with Fig. 3 and with additional brief information on CST and up-to-date new referencing. However, it is not informative without inclusion of the mechanisms of entire clusters with proper seismicity. Otherwise, this can be removed from the manuscript.

We agree that this figure is no longer necessary, so it is now removed. The coulomb stress modeling section is now updated with new references.

Fig. S7: This figure presents time domain waveform fits of selected P-and SH-body waves,and regional surface waves for the mainshock. However, the distribution of teleseismic stations are plotted without station codes in Fig. S2 which makes it difficult to analyse these closely. Thus, it is making even more difficult for readers to identify waveform fits. Besides, there is a huge azimuthal gap in the North and North-East quadrants spanning from Greenland to Kamchatka peninsula? It is NOT proper to present automated figure generations in the Grond software toolbox of Heimann et al. (2018). The authors should refine these graphics to be more relevant for the readers to orient themselves. Furthermore, brief introductory explanations should be provided in order to summarize the main features of the Grond software toolbox. Specifically, filtering is dangerous, and should be clarified properly reasons why. Otherwise, it looks like an output of the computing in a black-box fashion.
We are sorry the reviewer thinks the plots are not easy enough to read. Together with Fig. S2 we think we illustrate and report on the results to a full extent, as we say in the reply

regarding Fig. S2. While, of course, illustrating the results for large data sets is always difficult and assessing the results in detail will always take time. That the plots are generated automatically does not have to mean they are not valuable in our opinion. Please also see the reply for comment on Fig. S2.

The filtering is discussed in the main text and chosen accordingly to our target (section mainshock: lines 105 and 110), here namely teleseismic broadband data are inverted in the low-frequency range (0.01-0.05 Hz, now mentioned in the caption).

Please also see the main reply #5.

Figs. S8-S13: The authors should add colour bars for the misfit values. Also,it would be great to contribute some explanatory information regarding the optimization procedure used throughout in this article. For instance, I am not even sure what type of data you are displaying as misfits? Is this the total misfit obtained from the contribution of different data sets (strong motion, teleseismic, geodetic, etc.)? The detailed clarification is needed as this issue is rather critical.

The optimization results of finite-fault modelling (InSAR and strong-motion), point sources of the mainshock (teleseismic and regional data) and fore- and aftershocks (regional data) together with model parameters uncertainties, parameter trade-offs, the Grond input configurations, and detailed output reports are available in a separate data publication at: https://data.pyrocko.org/scratch/grond-reports/2020-elazig-sivrice/#/

Fig. S8:

Sequence plots of distribution and uncertainties of some parameters are a bit confusing one, and it does not help much with 68% confidence intervals.

Please see the reply for the comment of Figs. S8-S13.

Fig. S9:

Bootstrap misfit of the optimization is also too technical and does not help the readers much. Therefore, this figure should be removed.

Please see the reply for the comment of Figs. S8-S13.

Fig. S10:

Yes, I agree that MT decomposition is not well presented, and requires further analyses. Therefore, this figure should be removed.

Please see the reply for the comment of Figs. S8-S13.

Fig. S11:

Source parameter's scatter plots are not easily readable and does not help the readers much. Therefore, this figure should be removed especially when considered the azimuthal gaps of the broad-band stations used.

Please see the reply for the comment of Figs. S8-S13.

Fig. S12:

It refers to time domain waveform fits for strong motion data. Again, I do have reservations on this data-set and how the authors were able to get an access these data before the official release date of the 16 June 2020. So where is the doppler effects and directivity on strong-motion data? Also, it is NOT proper to present automated figure generations in the Grond software toolbox of Heimann et al. (2018). The authors should refine these graphics to be more relevant for the readers to orient themselves. Furthermore, brief introductory explanations should also be provided in order to summarize the main features of the Grond software toolbox. Specifically, filtering is dangerous, and should be clarified properly reasons why. Otherwise, it looks like an output of the computing in a black-box fashion.

See general comment #4. We have refined this figure together with strong motion station distribution.

The explanation about the method and Grond software has been added in the methodology section and with citation for a new published paper (Kühn et al., 2020).

However, we have completely reformulated the methodological section and we now provide an accurate description of the procedure to resolve point and finite source parameters.

*Kühn, D., Heimann, S., Isken, P. M., Ruigrok, E., Dost, B.: Probabilistic Moment Tensor Inversion for Hydrocarbon-Induced Seismicity in the Groningen Gas Field, The Netherlands, Part 1: Testing. Bulletin of the Seismological Society of America doi: https://doi.org/10.1785/0120200099, 2020.*

Fig. S13:

Finite Fault model plots of distribution and uncertainties of some source parameters of FF are a bit confusing one, and this figure does not help much to convince the reader especially when considered the azimuthal gaps of the broad-band stations used. Therefore, this figure should be removed.

Please see the reply for the comment of Figs. S8-S13.

Moreover we only use near-field data (Strong motion and InSAR) in finite Fault (FF) model, negher teleseismic nor regional broadband data.

Fig. S14:

Bootstrap misfit of the optimization for the FF model is also too technical and does not help the readers much. Therefore, this figure should be removed.

Please see the reply for the comment of Figs. S8-S13.

Fig. S15:

Source parameter's scatter plots for the FF model are not easily readable and does not help the readers much. Therefore, this figure should be removed especially when considered the azimuthal gaps of the broad-band stations used.

This figure shows the trade-offs between pairs of model parameters. Please see the reply for the comment of Figs. S8-S13.

Table S1: Moment tensor inversion results of the foreshocks and aftershocks are summarized in Table S1. However, I would like to see individual plots of complete waveform-fits of each earthquake spanning from 4 April 2009? right date? Otherwise, from 27 December 2019 to 19 March 2020. Are they regional moment tensor (RMT) results or else?

We apologize for the 4 April 2009 (typing error), which is 4 April 2019. The table reports our own regional MT solutions (we clarified this in the caption) and other solutions reported.
All detail of the results moment tensor inversion cannot be included in this manuscript for obvious space reasons.
we provide the detailed output reports for all inversion runs in a separate online report at:
https://data.pyrocko.org/scratch/grond-reports/2020-elazig-sivrice/#/

Moreover, the last access of the aftershocks was March 31, 2020 in the submitted manuscript. Now we have completed the aftershocks catalog (Last accessed 15 August 2020) and we obtain the focal mechanism solutions of 4 more events with magnitude larger than 4.3 which occurred between 31 March and 15 August 2020. In total we calculate the 21 focal mechanisms using the regional KOERI and GEOFON seismic networks.

How reliable are these mechanisms? How about the error bars in the earthquake mechanisms of both nodal planes? The waveform modelling for earthquake and tsunami source studies is a tedious profession and it takes longer time and careful consideration. Thus, I advise authors to be very careful at this kind of studies.

We thank the reviewer for this remark! As we routinely estimate model uncertainties we have this information for our own solutions. We compared our results with other available solutions. The average Kagan angle for the fore- aftershocks, when compared to reference solutions is ~30°.
See also the previous comment.

Referencing:
1. Acarel et al. (1996) paper is an irrelevant and poor one and not an objective good quality paper to cite as there are major misleading information included. Are you aware of them? Therefore, the adapted local crustal model is not valid and not reliable one to rely on further.

We assume the reviewer refers to Acarel et al. (2019) (not 1996). We have no reasons to consider this a "poor" or "non-objective" study, published in an ISI journal. The same velocity model has been used by Pousse-Beltran et al. (2020), now published in GRL.

2. I do not see quite relevance of the below articles besides being case studies. I advise removal of one of the below articles?

-Asayesh, B. M., Hamzeloo, H. and Zafarani, H.: Coulomb stress changes due to main earthquakes in Southeast Iran during 1981 to 2011. J Seismol 23, 135–150, https://doi.org/10.1007/s10950-018-9797-y, 2019.

-Asayesh, B. M., Zafarani, H., and Tatar, M.: Coulomb stress changes and secondary stress triggering during the 2003 (Mw 6.6) Bam (Iran) earthquake. Tectonophysics, 775, 228304, https://doi.org/10.1016/j.tecto.2019.228304, 2020.
We removed the first reference.

3. The statement in Lines of 60-64 is not true as Bulut et al. (2012) was not the first to report. "Bulut et al. (2012) characterize the EAF as a left-lateral strike-slip system,involving NE-SW and EW oriented segments which run parallel to the segmented trend of the main fault. Besides the dominant strike-slip mechanisms, Bulut et al. (2012) found evidence for additional thrust faulting on EW trending structures and normal faulting on NS trending secondary faults."
I advise authors carefully to read scholarly written papers on the Anatolian seismotectonics and geodynamics studies in order not to reach such strong conclusions. There are many sentences like these throughout the manuscript as authors are misusing cross-referencing, and therefore not giving the right credit who deserves much in the first place. I repeat here again that the authors need to invest some further reading sessions on the above topics.

We only state that Bulut et al. (2012) reported this, not that this would be the first study. However we have added the following sentences with references in the introduction to report the first studies:

*"The EAF was firstly named and described as a strike-slip transform fault by Arpat and Şaroğlu (1972), and has been the target of many seismological studies since."*

*Arpat, E., and F. Şaroğlu (1972), Some observations and thoughts on the East Anatolian fault, Bull. Miner. Res. Explor. Inst. Turk., 73, 44–50.*

4. It looks as if data-base of the main faults of Basili et. al. (2013) is NOT the right resources to make use of it? Subsequently, remove this reference and use a decent relevant one on the active faults of Turkey and surroundings. The neotectonics features of the Anatolia is well studied and established and is widely known. So why to refer to an incomplete data-base?
Basili, R., Kastelic, V., Demircioglu, M. B., Garcia Moreno D., et al.: The European Database of Seismogenic Faults (EDSF) compiled in the framework of the Project SHARE. http://diss.rm.ingv.it/share-edsf/, doi: 10.6092/INGV.ITSHARE-EDSF, 2013.
We could find no evidence in the literature to consider this database (European Database of Seismogenic Faults) incomplete, which is considered a reference for the European region.
http://diss.rm.ingv.it/share-edsf/

5. Line 131, the authors report that "Some surface cracks, rockfalls, landslides, and liquefaction were reported (Lekkas et al., 2020)".

Lekkas et al. (2020) did not execute field excursions after the mainshock in the area to map and to report such observations. This is not right referencing, and proves another example of wrong usage of cross-referencing! Check Turkish official report of MTA (2020) at the right

web page. Otherwise, one can easily form a paper simply navigating at the virtual space to get information in a copy and paste fashion.This is a serious issue and can be considered as a misconduct as decent piece of scholarly science requires sensitive and careful analyses ever. MTA. (2020). Preliminary field and evaluation report on 24 January 2020 Sivrice (Elazıg) Mw 6.8Earthquake, General Directorate of Mineral Research and Explorations of Turkey (MTA), Ministry of Energy and Natural Resources, Ankara, 48 pages (https://https://www.mta.gov.tr/).

We agree. We have replaced the suggested reference and we have added the clear sentences as the surface rupture was the concern of Rev#1. Furthermore we have added the new sentence which mentioned the recently published study and their results about the obscene of clear surface rupture (Pousse-Beltran et al. (2020)).

*"no significant surface rupture is reported by the General Directorate of Mineral Research and Explorations of Turkey (MTA, 2020) nor apparent in optical satellite imagery (Pousse-Beltran et al., 2020). There is a pronounced slip deficit above the mainshock rupture, leading to, if any, very weak fault motion in some parts of the fault's surface trace (Pousse-Beltran et al., 2020)"*

6. The authors should also consider large aftershocks observed striking NE-SW along the EAFZ before jumping on wrong conclusions with those of Nissen et al. (2019). Thus, what is the direct relevance of the below article in the current study? I would have written a serious comment on the below article, but I do not have much time to invest on this adventure.
Nissen, E., Ghods, A., Karas.zen, A., Elliott, J. R., Barnhart, W. D., Bergman, E. A., Hayes, G. P., Jamal-Reyhani, M., and et al.: The 12 November 2017 M w 7.3 Ezgeleh-Sarpolzahab (Iran) earthquake and active tectonics of the Lurestan arc. Journal of Geophysical Research: Solid Earth, 124. https://doi.org/10.1029/2018JB016221, 2019.

We assume the reviewer refers here to our sentence "… clear rupture directivity has not played a role in producing a larger number of aftershocks ahead of the main rupture direction as observed for other unilateral rupture earthquakes (Gomberg et al. 2003, Nissen et al. 2019)".
We agree that this sentence is not so important and is not a key sentence and simply can be removed with its references.

Data Availability:
I am quite curious how the authors obtained the unreleased AFAD's strong-motion data which should be clarified and confirmed in writing from the Turkish government authorities. Otherwise, this does NOT grant an equal opportunity on Data Availability for international scientists to conduct a research on the current and other relevant earthquakes in the region for global and/or regional mutual interests. Therefore, I consider this current work being NOT an objective piece of scientific conduct, and it is quite unfair to the others interested to

study these events further. Similarly, why did the authors NOT use any waveforms from the local and regional broadband stations operated by the AFAD?

See reply to main comment #4.

The AFAD regional broadband stations were not used for the point source inversion, because data were only available for the mainshock and few early aftershocks.

Software Availability:

Some of the tools are available for the broad scientific studies, but the details and decent expertise are rather limited. This issue should be enhanced in the text with right referencing, and note as SOM.

All used software is open source and includes user manuals and references. However, we have improved the description of used methodologies (see reply to main comment #1).

Language: I feel that the manuscript is rather unfocussed and could also clearly benefit from careful editing by native speaker as the written English needs some brushing up. I can point out several places where this needs to be done below, but certainly not every occurrence.

Line 206 reads "The mainshock started to nucleate from the topper part of the fault plane (Fig. 3b)". What does "topper" mean? Any good grammar? British/American English or a slang word invented?

We apologise for some typos; the manuscript has now been revised.

Discussions and Conclusions:

In addition, I would like to hear the authors' overall comments on the following submitted and accepted articles that I have recently acquired on the dedicated web pages.

A. I have just noticed the following accepted article on the 2020 Elazig earthquake, which is online since 29 March 2020 under URL (https://www.essoar.org/doi/10.1002/essoar.10502613.1), and I wonder how and why authors did not note/comment on this as they claim that they are jointly using many common available data-sets. Léa Pousse-Beltran et al. (2020). The 2020 M w 6.8 Elazıg (Turkey) earthquake reveals rupture behaviour of the East Anatolian Fault, AGU-GeophysicalResearchLetters(GRL), https://agupubs.onlinelibrary.wiley.com/doi/abs/10.1029/2020GL088136, also available at ESSOAr| https:/doi.org/10.1002/essoar.10502613.1, First posted online: Sunday 29 Mar 2020. Pousse-Beltan et al. (2020) also deal with the 2020 Mw 6.8 earthquake, and its rupture properties by using satellite geodesy and seismology. They mainly investigate the mainshock rupture, postseismic deformation and aftershocks, and relations to previous earthquakes. According to their model, to the ENE the mainshock may have propagated into the rupture zone of the 1874 M ~7.1 Golcuk Golu earthquake, and then stopped in the Lake Hazar basin, considered hosting a major EAF segment boundary. To the WSW the rupture propagated to the WSW at ~2 km/s and halted after~20 s along a straight, structurally simple section of the Puturge fault segment. Furthermore, their study indicates bilaterally

propagating rupture at relatively slow propagation speed from a nucleation point on an abrupt ~10 fault bend. Their model suggests the mainshock rupture with a pronounced shallow slip deficit that is only partially recovered through shallow afterslip and they keep discussing further. However, there is no significant surface rupture observed at distinctive studies already reported.

Hence, outstanding and open questions are:
1. I have further noticed by closely analysing InSAR data that this more complex geometry is NOT necessary to fit the InSAR observations as Pousse-Beltran et al. (2020) accomplished two disconnected fault planes with different dip to fit the InSAR data. So, I wonder what is the opinion and/or explanation of the authors on this matter? Explain it in details as you both use similar type of data-set in order to help reader of wider geological community.
2. What are the major discrepancies among their findings and major results in the present work?
3. How and why do they interpret the overall results by using both seismology and InSAR data?
4. The authors should add through discussion on this article at Discussion/Conclusion sections.

We recognised this recent paper only after our submission. We are glad to compare our results with those by Pousse-Beltran et al. (2020), now published online in GRL. In particular, we find that they also predict a unilateral rupture, not a bilateral one! Thus, there is a good agreement on this aspect of the rupture process, that they resolve with a different, back-projection approach not the same data-set as our study.
Rupture velocity, rupture duration, source depth, and other source parameters are also in very good agreement with our results, although obtained with different methods and partially different data.
We now cite more extensively their valuable works in our study.

B. Recently, Bletery et al. (2020) calculated a coupling map from InSAR and GNSS long-term velocities which suggests regions with slip deficit between 50-80% along the ruptured fault segment. Is there any further discussion and comments on this by the authors?
Bletery, Q., Cavalie, O., Nocquet, J-M., and Ragon, T. (2020). Distribution of interseismic coupling along the North and East Anatolian Faults inferred from InSAR and GPS data, submitted to AGU-Geophysical Research Letters, Earth and Space Science Open Archive (https://www.essoar.org/) Published Online: Thu, 5 Mar 2020, https://doi.org/10.1002/essoar.10502450.1.

Infact, we already cite this and other papers (see line 55 of our former manuscript): "Cavalié and Jónsson (2014) and Bletery et al. (2020) proposed heterogeneous and shallow (~5 km) locking depth for the EAF", which we discuss later in the discussion: "Our finite source model for the Elazığ-Sivrice earthquake suggests that this earthquake broke a shallow asperity, compatible with the shallow locking depth ..." and mention in the abstract.

This study is now published in the GRL.

Bletery, Q., Cavalie, O., Nocquet, J-M., and Ragon, T.: Distribution of interseismic coupling along the North and East Anatolian Faults inferred from InSAR and GPS data, Geophysical Research Letters, 47, e2020GL087775. https://doi.org/10.1029/2020GL087775, 2020.

C. I have also noticed the following article that refers to the 2020 Elazig earthquake, which is on line since 4 February 2020under URL (https://eartharxiv.org/8xa7j).Jonathan R. Weiss et al. (2020). High-resolution surface velocities and strain for Anatolia from Sentinel-1 InSAR and GNSS data. EarthArXiv Preprints, https://doi.org/10.31223/osf.io/8xa7j.

Weiss et al. (2020) claims that their "3D velocity and strain rate fields illuminate deformation patterns dominated by westward motion of Anatolia relative to Eurasia, localized strain accumulation along the North and East Anatolian Faults, and rapid vertical signals associated with anthropogenic activities and to a lesser extent extension across the grabens of western Anatolia".

I wonder how and why authors did not note/comment on this as they are also using assembled InSAR data-set in the Anatolia. Thus, I would like to hear what is the opinion and/or explanation of Jamalreyhani et al. on this matter? The authors should explain it in details as they both use InSAR data-set in order to help reader of wider geological community.

We are thankful to point out this paper. The study of Weiss et al. (2020) is based on a time-series analysis from InSAR data to resolve the long-term surface motion. With newer data they confirm observations at the EAF made earlier by Cavalie and Jonsson (2014) (see also answer to the previous comment). We added this reference to the statement given in the previous comment.

D. I wonder why the authors did not make use of the GNSS observations as they privilege(!) that they are using all the available data-set collected in the Anatolia.

We did not have direct access to coseismic surface offset measurements based on GNSS data. Such GNSS data provide pointwise 3D measurements of the surface motion in contrast to InSAR that provide Line-of-sight projections only. However, we combined ascending and descending InSAR data such that EW and vertical motion is well captured. In combination with seismological data that well cover all azimuths around the earthquake, we have enough observation to robustly estimate the parameters of shear dislocation as we show with our model uncertainty estimations. Additional GNSS data may contribute to lower the estimated model uncertainties, but would not have changed the picture at large, based on simulations that analysed the influence of a missing north-component in InSAR observations on the modelling of earthquake sources (InSAR Sensitivity Analysis of Tandem-L Mission for Modeling Volcanic and Seismic Deformation Sources by Ansari, Homa and Goel, Kanika and Parizzi, Alessandro and Sudhaus, Henriette and Adam, Nico and Eineder, Michael (2015) *InSAR Sensitivity Analysis of Tandem-L Mission for Modeling Volcanic and Seismic Deformation*

*Sources.* In: Proceedings of ESA (SP-371), SP-371, pp. 1-8. FRINGE 2015, 23-27 March 2015, Frascati (Rome), Italy. https://elib.dlr.de/94895/).

E. Furthermore, I would like to see the Finite-Fault Slip Distributions on the preferred fault plane mechanism of the authors by using individual data-set, in pairs and with entire data-set that the authors have. For example, local data (the strong motion, AFAD?), and regional seismology data (KOERI, AFAD, GEOFON?) and teleseismic body-wave inversions may recover zones of large slip, while they are combined into a single large zone in the slip distribution by the geodetic inversion (InSAR? or GPS?).

The authors have only provided Finite-Fault slip-inversion jointly using InSAR and a few strong motion data. I am puzzled to see that they have not used available teleseismic and regional data-set? Then, we may continue debating in discussions and making resolved conclusions.

A new figure is needed on Finite-Fault Slip Distribution integrating below data-set separately.

(a) Teleseismicbody waves (GDSN, FDSN, through IRIS DMC or else)
(b) Local seismic networks (AFAD, KOERI or else?)
(c) Regional seismic waveforms (AFAD, KOERI, GEOFON or else?)
(d) Strong Motion (AFAD, KOERI or else?)
(e) Geodetic (InSAR)(ESA, NASA or ALOS?)
(f) Coulomb (Cautious tidies work should be conducted)
(g) Seismicity (AFAD and/or KOERI?)
(h) Joint inversion with any of the above data-set to compare with each other.
(i) Full Inversion of all the above data-set.

I would like to see grid-space along-strike and along-dip with finite-fault slip distribution on these cells delineated with slip-vectors and displacement values (e.g. D-maximum, D-average), and evolution of seismic moment release as a function of time (i.e., source time function).This must not be too difficult to resolve and to retrieve over the inversion tools as there are much data.

With the single data sets alone the non-linear modelling of the finite source is not meaningful and would be very ill-posed. Using both ascending and descending InSAR sets alone, we would only be able to infer the static parameters of the source. Also, InSAR data, as near-field data, loses resolution for slip at larger depth. Such a model would look different, of course, from the combined data model we present. InSAR is important for finite-fault inversion to fix the lower-frequency image of the source and provides spatial resolution and constrains the fault position (Ide, 2007). Seismic data, especially near-field strong motion data is essential to resolve the temporal change in detail and provide better resolution (Anderson, 2003: Ide, 2007). The InSAR data can be joined with a near-field seismic data (High frequency strong motion) to better constrain the total slip, which is not well constrained by seismic data set alone (Ide, 2007). Using joint inversion for the finite fault

modeling has been used in a number of recent studies in other regions (Delouis et al. 2002; Zhang et al. 2012; Cesca et al. 2017; Gombert et al. 2019).

*Anderson, J. G.: Strong-motion seismology. International Geophysics Series, 81(B), 937–966, 2003.*

*Cesca, S., Zhang, Y., Mouslopoulou, V., Wang, R., Saul, J., Savage, M., Heimann, S., Kufner, S.-K., Oncken, O., Dahm, T. (2017): Complex rupture process of the Mw 7.8, 2016, Kaikoura earthquake, New Zealand, and its aftershock sequence. - Earth and Planetary Science Letters, 478, pp. 110—120.*

*Delouis, B., Giardini, D., Lundgren, P., and Salichon., J., 2002, Joint Inversion of InSAR, GPS, Teleseismic, and Strong-Motion Data for the Spatial and Temporal Distribution of Earthquake Slip: Application to the 1999 Izmit Mainshock., Bulletin of the Seismological Society of America, 92, 1, pp. 278–299.*

*Gombert, B., Duputel, Z., Shabani, E., Rivera, L., Jolivet, R., & Hollingsworth, J. (2019). Impulsive source of the 2017 $M_W$=7.3 Ezgeleh, Iran, earthquake. Geophysical Research Letters, 46, 5207– 5216. https://doi.org/10.1029/2018GL081794.*

*Zhang, Y., Feng, W.P., Chen, Y.T., Xu, L.S., Li, Z.H., Forrest, D., 2012. The 2009 L'Aquila Mw 6.3 earthquake: a new technique to locate the hypocentre in the joint inversion of earthquake rupture process. Geophys. J. Int. 191, 1417–1426. http://dx.doi.org/10.1111/j.1365-246X.2012.05694.x*

Using the regional seismological data alone or teleseismic data alone for finite fault inversion, we would miss a lot of information as well. We use low-frequency signals (regional and teleseismic broadband data) for the moment tensor inversion (Point source approximation).

Recent studies (e.g. Steinberg et al, 2020) show that near-field data (e.g. InSAR) are generally more sensitive to rupture segmentation of shallow earthquakes than far-field data (e.g. teleseismic data).

We have improved our methodological description as well as mainshock section to make more clear our procedure and to clearly state we invert for a homogeneous slip distribution.

*Steinberg, A., Sudhaus, H., Heimann, S., Krüger, F.: Sensitivity of InSAR and teleseismic observations to earthquake rupture segmentation, Geophysical Journal International, https://doi.org/10.1093/gji/ggaa351, 2020.*

F. The authors then can plot map-view of any of the above preferred ones on the morphology in order to compliment neotectonic and seismotectonics maps. Afterwards, we may then continue debating in discussions and making stable conclusions for likely future earthquakes in the region.

See previous reply. The slip distribution is homogeneous. Said that, we believe our conclusions on the future earthquake scenarios are discussed on the basis of robust results.

Best regards,
Mohammadreza Jamalreyhani
(on behalf of all co-authors)

---

## Author Comment (AC3) · 9 Sep 2020

**Reply Anonymous Referee #3**

We are grateful to the reviewer 3 for many valuable comments and suggestions and positive feedback. Below, we report answers to all reviewer comments. The original review text is in black, our reply in green.

General Comments:

The coseismic data from some seismological networks and from SAR Sentinel-1 satellite are analyzed in order to estimate the fault parameters of the 24 January 2020 earth-quake, understand the aftershock distribution, and the future distribution of events on the EAF. The paper is well structured and written. It represents an interesting application of mature software, with some interesting conclusions about the seismic gaps on the EAF fault. But, some conclusions and discussions are not examined with sufficient details, and some sentences are not completely debated. The time correlation among the seismic events can be not studied (only) with an elastic model (Coulomb 3.3), but using also other types of models, for example, visco-elastic, visco-plastic. Some connection between the probable forecast events and the mainshock should be discussed with more detail, especially for the journal where the authors have submitted. The reviewer suggests acceptance after major revision.

We are thankful for the overall appreciation of our research. We have carefully considered all suggestions and discussed them below, providing specific details on how they were accounted to improve our manuscript.

Scientific Comments:

In the Introduction, the authors describe briefly the geodynamic context about the Anatolian plate and the East Anatolian Fault. The slab pull model and mantle flow model are only two of the several models discussed in the literature. For, example, the lateral extrusion of crustal wedges as discussed in Mantovani et al. 2001 (Short and long term deformation patterns in the Aegean-Anatolian systems: insights from space geodetic data (GPS) and Numerical simulation of the observed strain field in the central-eastern Mediterranean region) explain the kinematic of the Anatolian plate using a different point of view. I think, for the sake of completeness it is right to describe briefly and mention the other models of the Mediterranean geodynamic pattern.

Following the reviewer's suggestion, we extended the introduction with respect to the regional geodynamics and included the suggested reference. This surely helps to provide a broader view on the different models proposed in the literature. Our study and results are possibly too specific (i.e targeting a single seismic sequence) to contribute to the discussion on these different hypotheses. We have added the following sentences in the introduction:

*"Mantovani et al. (2001) have further hypothesized a role of the post-seismic relaxation induced by seismic activation of the NAF in the current kinematics of the Anatolian block."*

The paper represents an interesting application of mature software to analyze and inversion of seismic and SAR/GNSS data. Also, the authors use the Coulomb 3.3 software in order to estimate the coseismic static stress changes. The authors have developed and elastic model in order to estimate the spatial evolution of the Coulomb stress and they have discussed the correlation between the stress pattern and aftershocks distribution. Also, they have suggested that the increased stress in some parts of the EAF can expedite large earthquake activity in this region. I think that this elastic approach is a good model to understand the aftershock distribution, but to study the time distribution of the seismic events in an area it is necessary to use other models, for example, a visco-elastic model where the visco-elastic proprieties of the lower crust can be modeled and reproduce the time evolution of the stress field in the study area. I suggest to the authors introduce in the discussion and/or conclusion paragraph a brief discussion about the problems and limitations of the elastic model when are used in the earthquake correlation time studies.

Thank you for the comment. We agree that the Coulomb stress analysis we perform and discuss has strong limitations, and indeed only discuss it in terms of the spatial distribution of the aftershocks. Using other models, accounting e.g. for viscoelasticity, is beyond the scope of our paper. We accounted for this suggestion by adding new paragraphs in the "Coulomb stress" and "discussion" sections, stating the limitations of the elastic model. Please see the related comments in the technical corrections.

Technical corrections:

Line 44:….it did not host major earthquakes during the last hundred years (Fig. 1): the most recent, large earthquake on the EAF dates to 1971……A more strong earthquake along EAF was 2010…..The 1971 event has occurred only about 50 years ago, and 2010 is only a few 'geological seconds' before now. It is not clear why the authors speak about the last hundred years. I agree with the authors that the large earthquake recurrence time on the

EAF is greater than the NAF, but I suggest to the authors to modify the time span in these sentences in order to have an agreement.

We apologize for the misunderstanding and thanks for pointing out the lack of clarity. We changed "large" to "strong" according to the widely used earthquake classification based on Stein and Wysession (2003).

| Class | Magnitude |
|-------|-----------|
| Great | 8 or more |
| Major | 7 - 7.9 |
| Strong | 6 - 6.9 |
| Moderate | 5 - 5.9 |
| Light | 4 - 4.9 |
| Minor | 3 - 3.9 |

Stein, S., & Wysession, M. (2003). An introduction to seismology, earthquakes, and earth structure.

To be clear within the manuscript, we have added a new sentence for a definition: ... *"it did not host major earthquakes during the last hundred years (Fig. 1): the most recent, strong earthquakes on the EAF date to 1971 (Mw 6.7 Bingöl earthquake, Duman and Emre,2013) and 2010 Mw 6.1 Kovancilar earthquake."*

Line 55: I think it is not completely correct to mention a paper only submitted.
Bletery, Q., Cavalie, O., Nocquet, J-M., and Ragon, T.: Distribution of interseismic coupling along the North and East Anatolian Faults inferred from InSAR and GPS data, Geophys.Res.Lett.Earth and Space Science Open Archive,https://www.essoar.org/doi/10.1002/essoar.10502450.1, submitted, 2020.

The Bletary et al. (2020) study is published now in the Geophysical Research Letters. We have edited the reference list as below:

*Bletery, Q., Cavalié, O., Nocquet, J. M., & Ragon, T. (2020). Distribution of interseismic coupling along the North and East Anatolian Faults inferred from InSAR and GPS data. Geophysical Research Letters, 47, e2020GL087775.* [https://doi.org/10.1029/2020GL087775](https://doi.org/10.1029/2020GL087775)

Line 69. Same consideration about a submitted paper. I think the mentioned results can be not reported.

Please see the previous comment.

Line 76: I suggest to the authors to use the same decimal digits about the Elazig-Sivrice earthquake (6.8) unless they have estimated the magnitude with associated uncertainty on the second decimal digit.

Thank you for pointing out the indeed too high precision we gave, also in your next comment.

Line 116: Unfortunately, I am a physics, and if I write 6.77±0.1 I do not pass the first exam of the Laboratory. I suggest to the authors to write 6.8±0.1 and change in the text substituting 6.8 at 6.77. In Table 1, about this study are reported two 6.7 values, perhaps these values are 6.8.

We fully agree, and we changed all numbers accordingly (e.g. 6.8±0.1), as suggested (also in the title). We fixed the wrong number in table 1.

Line 158: Most of the foreshocks including two Mw~5 are located very close to the mainshock nucleation point, suggesting that they could have played a role in the main-shock preparation. This is a 'strong' sentence with support of only two references, but it can have important fallout, why the authors believe these earthquakes could have a role in the mainshock preparation, these events have anticipated or delayed the main-shock?

"Thank you for pointing out this. We agree that only the vicinity of these earlier earthquakes to the nucleation point of the mainshock strictly does not suffice to suggest that they influenced the mainshock. More investigations would be needed to gain more indications for or against this hypothesis, which are beyond the scope of the study. We removed this sentence."

Line 187: Why do you use these values for Young, shear, and Poisson modulus?

Poisson's ratio (PR) changes between -1 to 0.5; 0.25 is typically used in the laboratory tests and most crustal rocks have Poisson's ratio between 0.25 and 0.3. For Young's modulus (E), $8 \times 10^4$ MPa is typically used for rocks that are located in this range of depths. So based on the G = E/[2(1+PR)] relation, the shear modulus (G) is equal to $3.2 \times 10^4$ MPa.

Line188: I suggest to the authors to discuss briefly why they have chosen the middle value of the apparent coefficient of friction.

We have added these sentences to the "Coulomb failure stress change analysis" section:

*"Choosing an appropriate value for the coefficient of fault friction is difficult. Based on laboratory experiments on the frictional slip within rocks, the apparent coefficient of friction has high values such as 0.5-0.8 (e.g. Byerlee and Brace, 1968). On the other hand, increased pore fluid pressure due to injected fluid decreases the coefficient of friction. A high coefficient of friction (~0.8) is usually assumed for continental thrust faults, while for young and normal faults it may be higher. Low friction has been found to fit best for creeping faults, and very low friction (<0.2) for major transforms, such as the San Andreas (Harris and Simpson, 1998; Parsons et al., 1999; Toda and Stein, 2002). For strike-slip on unknown faults intermediate friction values of 0.4 are usually assumed (King et al. 1994; Parsons et al., 1999) and we follow these examples."*

*Byerlee, J. D., & Brace, W. F. (1968). Stick slip, stable sliding, and earthquakes—effect of rock type, pressure, strain rate, and stiffness. Journal of Geophysical Research, 73(18), 6031-6037.*

*Harris, R. A., & Simpson, R. W. (1998). Suppression of large earthquakes by stress shadows: A comparison of Coulomb and rate-and-state failure. Journal of Geophysical Research: Solid Earth, 103(B10), 24439-24451.*

*Toda, S., & Stein, R. S. (2002). Response of the San Andreas fault to the 1983 Coalinga-Nuñez earthquakes: An application of interaction-based probabilities for Parkfield. Journal of Geophysical Research: Solid Earth, 107(B6), ESE-6.*

*Parsons, T., Stein, R. S., Simpson, R. W., and Reasenberg, P. A.: Stress sensitivity of fault seismicity: A comparison between limited-offset oblique and major strike-slip faults. Journal of Geophysical Research: Solid Earth, 104 (B9), 20183-20202, 1999.*

*King, G. C., Stein, R. S., and Lin, J.: Static stress changes and the triggering of earthquakes. Bulletin of the Seismological Society of America, 84(3), 935-953, 1994.*

Line 192: In the caption of Figure S6 change Figure 3 with Figure 4 (I think).

Fixed.

Line 218: These loaded stresses can expedite future large earthquakes on either one of these segments..... I think that the Coulomb stress has been estimated on the fault plane of the previous earthquake or .... I suggest to the authors to explain in more detail these concepts.

We consider both segments in NE and SW ends of the ruptured plane as a receiver fault. Both segments are loaded by positive Coulomb stress change. Both parts experienced strong earthquakes in historical times (former Fig. 1).

We have decided to remove this sentence due to the limitation of the Coulomb stress modelling.

"These loaded stresses can expedite future large earthquakes on either one of these segments"

Line 227: I suggest to the authors to indicate the Figure where the aftershocks cloud north of the EAF can be seen. I think it is the cloud at the NE near the lake.

Figure 3 already shows this. We now have plotted this figure with increasing the thickness of EAF in the map and some other changes including the update of the later aftershocks (see the next comment). We also have referred to figure 3 explicitly in the text:

"The location of almost all aftershocks north of the fault trace confirms the fault plane dip towards NNW (Fig. 5)."

See the next comment for the updated former figure 3 (Fig. 5 in the new version).

Line 255: I suggest to the authors to report the three sectors discussed in Figure 3 in order to help the reader.

Thank you for the suggestion. We have shown the 3 sectors in the updated Figure. Moreover, we updated the aftershocks catalog (Last accessed 15 August 2020) and we calculated the focal mechanism of 4 more events with a magnitude larger than 4.3 which occurred between 31 March and 15 August 2020. All now include in former figure 3.

[Figure]

*Former Fig. 3 (Fig. 5 In the new version). Spatiotemporal evolution of the 2020 Mw 6.8 Elazığ-Sivrice earthquake sequence (black stars always denote the mainshock, purple and cyan circles show aftershocks and filled brown circles show foreshocks). a) Spatial distribution of seismicity at the Pütürge segment, located between the Hazar Lake and the Yarpuzlu bend, showing the path of Firat River (blue line). Red lines show main faults (after Basili et al., 2013). Circles represent the epicentral locations of fore- and aftershocks (purple circles show 18 days of relocated aftershocks from Melgar et al., 2020 and cyan circles show AFAD catalog Ml 1+ and azimuthal gap less than 120°, last accessed 15 August 2020). The grey filled box shows the surface projection of the modeled source, with the thick-lined edge marking the upper fault edge. Focal mechanisms of the mainshock, 2 foreshocks and 19 aftershocks (focal spheres, color scale according to centroid depths) shown based on our moment tensor inversion. Black squares denote locations of the closest strong motion stations with their code. b) Depth cross-section along the profile DD' of relocated aftershocks (after Melgar et al., 2020) and events larger than Ml 4 (cyan) from AFAD catalog. The light pink rectangle shows the main rupture area and the dark vector shows the direction of the main rupture propagation, as resolved in this study. (c1-3) Depth cross-sections along profiles AA', BB' and CC', respectively (dip and width of all cross-sections are 90° and 20 km, respectively), showing the focal mechanisms of largest events (cross section projection). d) Temporal evolution of the aftershocks (Ml 1+ and azimuthal gap < 120°) versus longitude; the upper histogram shows the longitude versus the number of events N. The light yellow patch covers the longitudes ruptured in the Elazığ-Sivrice based on our finite-source modelling. e) Temporal evolution of the foreshocks (same style as panel d).*

Line 264 and …: I can in agreement with the authors about the increasing of the stress on some fault segments due to the study earthquake. The problem could be represented that the elastic model adopted to give the 'instantaneous' stress increasing, as briefly discussed for the authors to provide the energy for the aftershocks. The possibility of a stress transfer could be investigated with viscoelastic or similar models where it is possible to model the distribution of the stress/strain in the time. But another approach could require a lot of time, therefore I suggest to authors to discuss briefly the different approaches between elastic and viscoelastic (for example) models and the kind of results that they can obtain.

We have added the following two paragraphs to the "Coulomb failure stress change analysis" sections:

*"In the last decades, many studies confirmed the role of earthquake interactions by reshaping the state of stress on nearby faults (Stein et al., 1992; Hainzl et al., 2010). The most notable mechanisms for explaining stress interaction between earthquakes are (Freed, 2006): (1) static stress transfer, due to permanent deformation in the vicinity of an earthquake's rupture surface (e.g. King et al., 1994; Steacy et al., 2005; Stein, 1999), (2) dynamic stress changes, due to the passage of seismic waves (e.g. Kilb et al., 2000; Felzer et al., 2006), (3) viscoelastic relaxation, and poroelastic rebound (e.g. Freed et al., 2001), (4) release of fluids during and after faulting, and fluid pore diffusion (e.g. Sibson et al., 1975; Sibson, 1981; Hickman et al., 1995). Among these, especially static stress transfer is widely accepted to play a key role (e.g., Stein et al., 1992; King et al., 1994; Harris et al., 1995; Yadav et al., 2012; Mitsakaki et al., 2013)."*

*And:*

*"In this study we use the software Coulomb 3.3 (Lin and Stein, 2004; Toda et al., 2005) to calculate the coseismic static stress changes. A uniform elastic half-space following Okada (1992) is assumed. The Coulomb Failure Stress Changes ($\Delta$CFS) are commonly used as a scalar indicator of the stress change. The change in CFS depends on shear stress change $\Delta\tau$, normal stress change $\Delta\sigma$, and the apparent coefficient of the fault friction on the receiver fault µ', which includes the unknown hydrostatic pore pressure change (King et al., 1994).*

$$\Delta CFS = \Delta\tau + \mu'\Delta\sigma \qquad\qquad (1)$$

*The large uncertainties are associated with the Coulomb failure stress change calculation, which are mainly due to non-unique slip inversions (Hainzl et al., 2009, Woessner et al., 2012), secondary stress triggering (Helmstetter et al., 2005), and poor knowledge about assumed receiver faults (Sharma et al., 2020). Here the uncertainties of the slip inversion are neglected in the calculation of $\Delta$CFS. Another limitation of the coseismic $\Delta$CFS model is that only elastic responses to fault slip are considered, and thus viscous effects in the triggering process remain unaccounted for (Freed and Lin, 2001; Freed, 2005). In the future it would be useful to calculate postseismic stress changes from viscoelastic stress relaxation by considering plastic models for the Earth. Nevertheless, first order effects of static stress changes are well represented with our simple approach."*

*Steacy, S., Gomberg, J., & Cocco, M. (2005). Introduction to special section: Stress transfer, earthquake triggering, and time-dependent seismic hazard. Journal of Geophysical Research: Solid Earth, 110(B5).*

*Kilb, D., Gomberg, J., & Bodin, P. (2000). Triggering of earthquake aftershocks by dynamic stresses. Nature, 408(6812), 570-574.*

*Felzer, K. R., & Brodsky, E. E. (2006). Decay of aftershock density with distance indicates triggering by dynamic stress. Nature, 441(7094), 735-738.*

*Freed, A. M., & Lin, J. (2001). Delayed triggering of the 1999 Hector Mine earthquake by viscoelastic stress transfer. Nature, 411(6834), 180-183.*

*Freed, A. M. (2005). Earthquakes triggering by static, dynamic, and postseismic stress transfer. Annu. Rev. Earth Planet. Sci., 33, 335-367.*

*Sibson, R. H. (1981). Fluid flow accompanying faulting: field evidence and models. Earthquake prediction: an international review, 4, 593-603.*

*Sibson, R. H., Moore, J. M. M., & Rankin, A. H. (1975). Seismic pumping—a hydrothermal fluid transport mechanism. Journal of the Geological Society, 131(6), 653-659.*

*Hickman, S., Sibson, R., & Bruhn, R. (1995). Introduction to special section: Mechanical involvement of fluids in faulting. Journal of Geophysical Research: Solid Earth, 100(B7), 12831-12840.*

*Yadav, R. B. S., Gahalaut, V. K., Chopra, S., & Shan, B. (2012). Tectonic implications and seismicity triggering during the 2008 Baluchistan, Pakistan earthquake sequence. Journal of Asian Earth Sciences, 45, 167-178.*

*Mitsakaki, C., Rondoyanni, T., Anastasiou, D., Papazissi, K., Marinou, A., & Sakellariou, M. (2013). Static stress changes and fault interactions in Lefkada Island, Western Greece. Journal of Geodynamics, 67, 53-61.*

*Pollitz, F. F., & Sacks, I. S. (2002). Stress triggering of the 1999 Hector Mine earthquake by transient deformation following the 1992 Landers earthquake. Bulletin of the Seismological Society of America, 92(4), 1487-1496.*

*Hainzl, S., Enescu, B., Cocco, M., Woessner, J., Catalli, F., Wang, R., and Roth, F. (2009), Aftershock modeling based on uncertain stress calculations, J. Geophys. Res., 114, B05309, doi:10.1029/2008JB006011.*

*Helmstetter, A., Y. Y. Kagan, and D. D. Jackson (2005), Importance of small earthquakes for stress transfers and earthquake triggering, J. Geophys. Res., 110, B05S08, doi:10.1029/2004JB003286.*

Sharma, S., Hainzl, S., Zöller, G., and Holschneider, M.: Is Coulomb stress the best choice for aftershock forecasting?. Journal of Geophysical Research: Solid Earth, 125, e2020JB019553. https://doi.org/10.1029/2020JB019553, 2020.

Line 264 .... It is not clear in the text which scenario the authors believe it is more realistic (1, 2, or 4). Please clarify this point.

We assume the reviewer refers to line 246. Thanks for pointing out the lack of clarity. As we have mentioned in line 244 of the submitted manuscript: Both scenarios, 1 and 2 are plausible and not in contradiction with the data. We have also added a new paragraph about the segmentation in the discussion section.

*"The near-field data, in particular InSAR data which is more sensitive to rupture segmentation of shallow earthquakes (Steinberg et al., 2020), do not indicate a strong rupture segmentation and the modeled synthetic data fit well enough to the observation without doubling the degree of freedom by allowing for two segments instead of one. With more free parameters the fit to the modeled data is quite naturally increasing, but if the increase in data fit is significant enough would need an extra analysis, e.g. based on informational criteria by Steinberg et al., (2020)."*

*Steinberg, A., Sudhaus, H., Heimann, S., Krüger, F.: Sensitivity of InSAR and teleseismic observations to earthquake rupture segmentation, Geophysical Journal International, ggaa351, https://doi.org/10.1093/gji/ggaa351, 2020.*

Line 284: Local seismicity clusters, appearing months prior to the Elazig-Sivrice earthquake occurrence, probably track the slip instability onset. Probably I am in agreement with the authors, but they could briefly explain why these events have increased the stress on the Elazig-Sivrice fault. There is also a possibility that they have decreased the stress on the fault.

We discuss these seismicity clusters months prior to the Elazig-Sivrice earthquake. But as the previous comment also refers to this issue, we removed the part of the discussion about the slip instability onset and the role in the mainshock preparation. More investigations would be needed to gain more indications for or against this hypothesis, which are beyond the scope of the study.

Line 525 Caption Figure 1: lost references about the kinematic pattern shown in the left up corner of the figure.

Fixed.

Line 675: lost reference about the active faults (Basili et al. 2013).

Fixed.

Best regards,
Mohammadreza Jamalreyhani
(on behalf of all co-authors)